# Histone deacetylase 11 inhibition promotes breast cancer metastasis from lymph nodes

Patrick L. Leslie[1,5], Yvonne L. Chao[1,5], Yi-Hsuan Tsai[1], Subrata K. Ghosh[1], Alessandro Porrello[1], Amanda E.D. Van Swearingen[1], Emily B. Harrison [1], Brian C. Cooley[1,2], Joel S. Parker [1,3], Lisa A. Carey[1,4] & Chad V. Pecot [1,4]

Lymph node (LN) metastases correspond with a worse prognosis in nearly all cancers, yet the occurrence of cancer spreading from LNs remains controversial. Additionally, the mechanisms explaining how cancers survive and exit LNs are largely unknown. Here, we show that breast cancer patients frequently have LN metastases that closely resemble distant metastases. In addition, using a microsurgical model, we show how LN metastasis development and dissemination is regulated by the expression of a chromatin modifier, histone deacetylase 11 (HDAC11). Genetic and pharmacologic blockade of HDAC11 decreases LN tumor growth, yet substantially increases migration and distant metastasis formation. Collectively, we reveal a mechanism explaining how HDAC11 plasticity promotes breast cancer growth as well as dissemination from LNs and suggest caution with the use of HDAC inhibitors.

---

[1] UNC Lineberger Comprehensive Cancer Center, University of North Carolina at Chapel Hill, Chapel Hill, NC 27599, USA. [2] Department of Pathology and Laboratory Medicine, University of North Carolina at Chapel Hill, Chapel Hill, NC 27599, USA. [3] Department of Genetics, University of North Carolina at Chapel Hill, Chapel Hill, NC 27599, USA. [4] Division of Hematology and Oncology, University of North Carolina at Chapel Hill, Chapel Hill, NC 27599, USA. [5] These authors contributed equally: Patrick L. Leslie, Yvonne L. Chao. Correspondence and requests for materials should be addressed to C.V.P. (email: pecot@email.unc.edu)

Metastasis causes ~90% of all cancer-related deaths. Although the most common site of initial spread in most cancers is to lymph nodes (LNs), which corresponds with a worsened prognosis, the contribution that locoregional LN metastasis has on seeding distant organs remains highly controversial[1–3]. For example, several large cohort studies have shown that the presence of microscopic LN metastasis corresponds with poor survival[4–6]. However, clinical trials investigating the relationship of LN metastasis treatment with survival have yielded conflicting results[7–9]. Furthermore, while molecular profiling and phylogenetic analyses suggest LN metastases often give rise to distant metastases in colon cancer (~35% incidence)[10], another study in breast cancer found no evidence that LN metastases are required for dissemination to distant sites[11]. In mouse models, several mechanisms promoting tumor lymphangiogenesis have also correlated with the development of distant metastasis[12–15]. Recently, the first direct experimental evidence of LN metastases seeding distant metastases has emerged[16–19], which showed that cancer cells egress from LNs predominantly through the draining LN blood vessels. However, the mechanistic underpinnings explaining how tumor cells exit the LN are still unknown. This represents a critical knowledge gap in metastatic biology[20], especially because the mechanisms governing LN metastasis may be unique from those promoting direct hematogenous spread from the primary lesion.

Here we show that LN metastases in breast cancer patients often phylogenetically resemble distant metastases. Using several experimental models, we show how LN metastasis establishment occurs through increased expression of a poorly understood chromatin modifier, histone deacetylase 11 (HDAC11). Interestingly, while genetic and pharmacologic blockade of HDAC11 decreases LN tumor growth, it substantially increases migration and promotes distant metastasis formation. Our findings demonstrate that establishment of LN metastasis, and then egress from LNs to distant sites, is both highly efficient and dynamic. Additionally, these findings reveal a next context for evaluation of cancer therapeutics and suggest caution with the use of HDAC inhibitors (HDACis).

## Results

**LN and distant metastases share a common origin.** Because a recent report found no evidence that LN metastases were required for the development of distant metastases in breast cancer[11], we first analyzed a cohort of seven breast cancer patients from the University of North Carolina Breast Cancer Rapid Autopsy Program (UNC RAP) for whom primary tumors (T), LN metastasis (L), and distant metastases (D) were collected[21]. Single-nucleotide variant data analysis with pairwise distance measurements (Jensen–Shannon distances (JSDs))[10] was used to determine whether the LN samples were phylogenetically closer to the primary lesion or to distant metastases. If in a patient's phylogenetic tree we observed that at least one distant metastasis sample was in the same clade with a LN sample and no other primary sample, we then concluded it as "LN-met mediated," which implies that the LN metastases gave rise to one or more distant metastases. In all other cases, we concluded it as "LN-met independent", which implies that the primary tumor directly seeded the distant site hematogenously. Compiling a ratio of the phylogenetic distance between D and L tumors over the phylogenetic distance between D and T tumors revealed that five out of the seven (71%) patients displayed evidence of a LN-met-mediated spread with at least one distant metastasis (Fig. 1a, b, d–f), suggesting that distant breast cancer metastases can be seeded from LN metastases in breast cancer. Bootstrapping analysis revealed high confidence classification of these clades

(Fig. 1c). In particular, distance ratios for patients A15, A20, and A34 showed that in some patients there is evidence that distant metastases in these patients share a more recent common ancestor with LN metastases than with primary tumors (Fig. 1a, b, d–f). These results suggest that LN metastases seeded distant metastases; however, we cannot exclude the possibility that the distant metastases instead seeded the LN metastases. We also found evidence for direct hematogenous route of metastasis from the primary tumor to distant sites, as we also observed in nearly all patients evidence of distant lesions that were more closely related to the primary tumors. Two out of seven patients displayed only "LN-met-independent" spread (Fig. 1c, g). These findings suggest that both lymphatic and direct hematogenous routes of spread commonly co-occur in breast cancer.

**LN metastases can give rise to distant metastases.** Despite recent advances in modeling how breast cancer spreads through LN blood vessels[18,19], the mechanisms and potential pharmacological targets that are involved remain unknown. Much of the complexity stems from the highly interconnected circulatory patterns of the lymphatic and hematogenous vasculatures[22,23]. To address this issue, we utilized a syngeneic mouse breast cancer cell line (4T1) capable of high-fidelity spontaneous metastases to LNs and distant organs[24]. While the 4T1 model histopathologically resembles triple-negative breast cancer, it also resembles the luminal molecular subtype[25]. Starting with the same parental 4T1 line, we generated dual fluorescence and luciferase reporter lines stably expressing either GFP/firefly luciferase (4T1-G/fL) or mCherry/Renilla luciferase (4T1-mCh/rL) as a means to analyze metastasis kinetics. To directly study cells growing in the context of the LN microenvironment, we developed a micro-injection model (Supplementary Fig. 1a). This approach allowed us to fine-tune direct injection into the draining axillary LN (AxLN) and consistently led to 68% take rates (Supplementary Fig. 1b). These AxLN tumors grew within the LN capsule similar to spontaneous AxLN metastases from orthotopic mammary fat pad (MFP) tumors (Supplementary Fig. 1a, c–e), and showed characteristics of natural tumor progression, including necrotic regions and metastasis to the lungs (Supplementary Fig. 1f, g). Using this LN micro-injection model, we determined the kinetics of distant metastasis to the lung using flow cytometry by analyzing various timepoints from 1 day to 6 weeks after AxLN micro-injection (Supplementary Fig. 2a, b). We found that distant metastases were readily detected in the lungs by 2 weeks post injection (Supplementary Fig. 2a).

**Dissemination by the lymphatic route is highly efficient.** Considering LNs drain to the lungs via efferent lymphatics or direct LN blood vessel invasion[18,19], we compared the metastases kinetics from LNs with direct hematogenous seeding by tail vein injection. Surprisingly, we found that LN tumors were significantly more efficient at generating distant metastases based on both number and frequency of micro-metastases in the lungs and brain (Supplementary Fig. 2c–f). To compare the metastatic efficiency of LN tumors with that of primary tumors, we injected equal cell numbers of 4T1-G/fL into the MFP and 4T1-mCh/rL into the corresponding right AxLN of the same mouse (Fig. 2a). The growth of each tumor could be differentiated based on luciferase signal, and tumor growth rate was similar between the two cell lines (Fig. 2b, c). After 6 weeks of growth, the vast majority of the AxLN tumors were mCh+, and <1% of tumor cells were GFP+, suggesting that distant 4T1-GFP/fL metastases most likely occurred via the hematogenous route (Fig. 2d). Using this model, we found that AxLN tumors metastasized to the lungs more efficiently than orthotopic MFP tumors, as indicated by

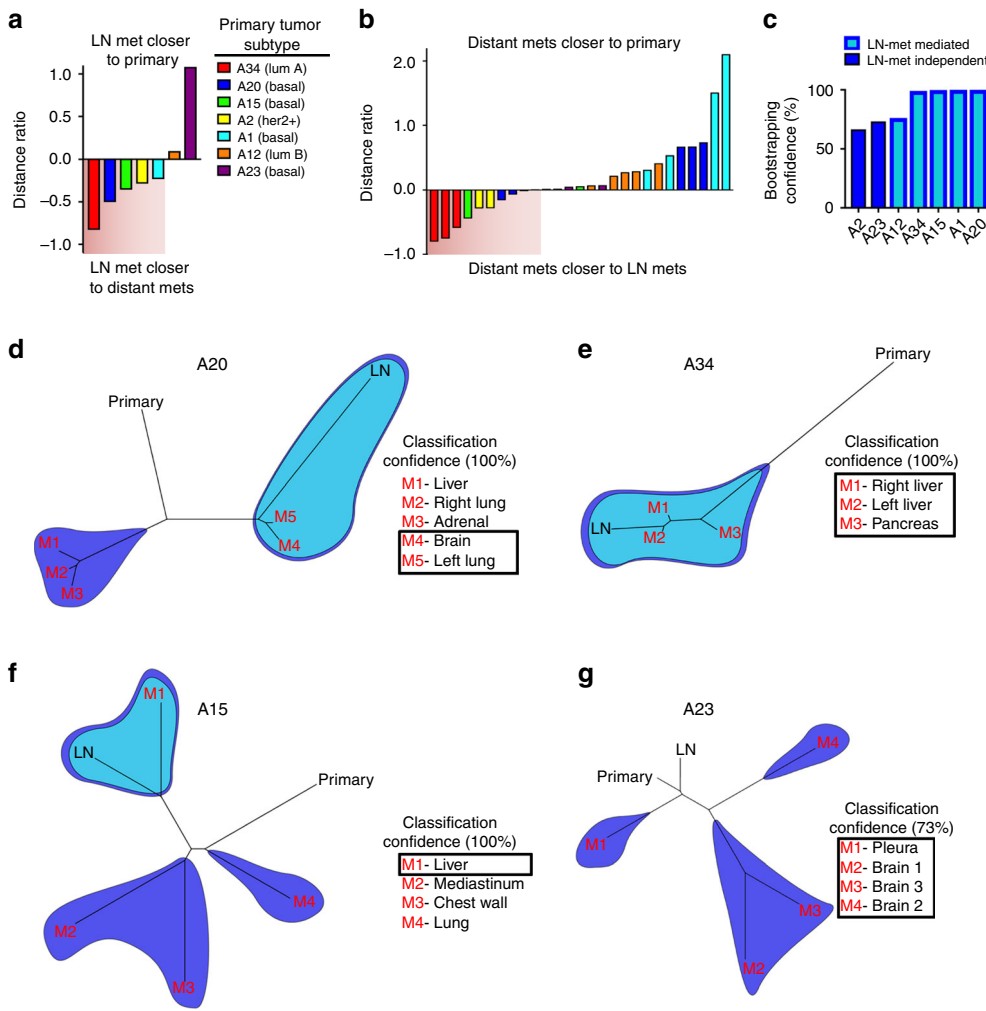

**Fig. 1** Metastatic breast cancer patient phylogenetic analysis show that LN metastases efficiently metastasize to distant organs. **a** Distance ratios from metastatic breast cancer patient samples on a patient and **b** individual metastasis basis. **c** Bootstrap confidence estimates for each group of patient samples. **d**–**f** Representative patient phylogenetic trees showing examples of LN-met- mediated and **g** LN-met-independent patterns of spread. Light blue shading demarcates LN-containing clades, whereas dark blue shading demarcates distant metastasis clades. Overlapping shading indicates clades containing both LN and distant metastases, implicating a LN-mediated pattern of distant metastases. Boxed in metastasis are those displaying a LN-mediated pattern of spread

both fluorescence and luciferase signal (Fig. 2e, f). To evaluate whether the difference in metastatic efficiencies was due to an intrinsic difference between the two reporter cell lines, we switched the orientation, micro-injecting 4T1-GFP/fL into the AxLN and 4T1-mCh/rL into MFP. Again, 4T1-GFP/fL cells injected into AxLN were more capable of establishing distant metastases in the lung compared to 4T1-mCh/rL cells injected in MFP (Supplementary Fig. 3a). To better delineate the contribution of hematogenous vs. lymphatic dissemination, we also compared injection of 4T1-mCh/rL cells into either the MFP, the MFP with AxLNs removed prior to injection, or into the AxLN. Compared with MFP-injected mice that had AxLNs removed, metastasis to lungs was significantly increased in the AxLN injection group (Supplementary Fig. 3b). There was no significant difference in lung metastases (LuMs) between the MFP and the MFP with AxLN removed groups. Taken together, these data show that distant metastasis occurs via both hematogenous and lymphatic seeding, and although hematogenous dissemination is likely the predominant route of spread, metastasis via LNs is more efficient.

**An epigenetic program regulates metastases via the LNs**. To determine how cancer cells spread from LNs, we isolated several

4T1 sub-clones from MFP- and AxLN-injected tumors, and LuMs that arose from micro-injected AxLNs (AxLN-LuM). After normalizing to 4T1-mCh/rL and 4T1-GFP/fL parental lines, we analyzed differentially expressed genes between MFP and AxLN sub-clones (Fig. 3a), and then between AxLN and AxLN-LuM sub-clones (Fig. 3b). Based on our analyses, we identified 206 genes that were differentially up- or down-regulated in at least one of these three microenvironments (Fig. 3c, Supplementary Fig. 4; see Methods for screening criteria). Gene ontology (GO) analyses revealed that the genes differentially expression in AxLN were predominantly involved in cell cycle progression (Fig. 3f). Consistent with a cell growth phenotype, we found that AxLN sub-clones divided substantially faster than MFP, AxLN-LuM, or LuM sub-clones derived from tail vein injections, suggesting an important role for proliferation within the LN (Fig. 3g). Among the proliferation regulators, *RRM2* and *E2F8* were the most differentially expressed genes (Fig. 3h). Moreover, of the differentially expressed genes, 152 (74%) were only up- or down-regulated in AxLN sub-clones, yielding two predominant patterns of differential expression between MFP, AxLN, and AxLN-LuM sub-clones: down-up-down or up-down-up (Fig. 3d, e). Based on these patterns of plasticity in gene expression, we hypothesized

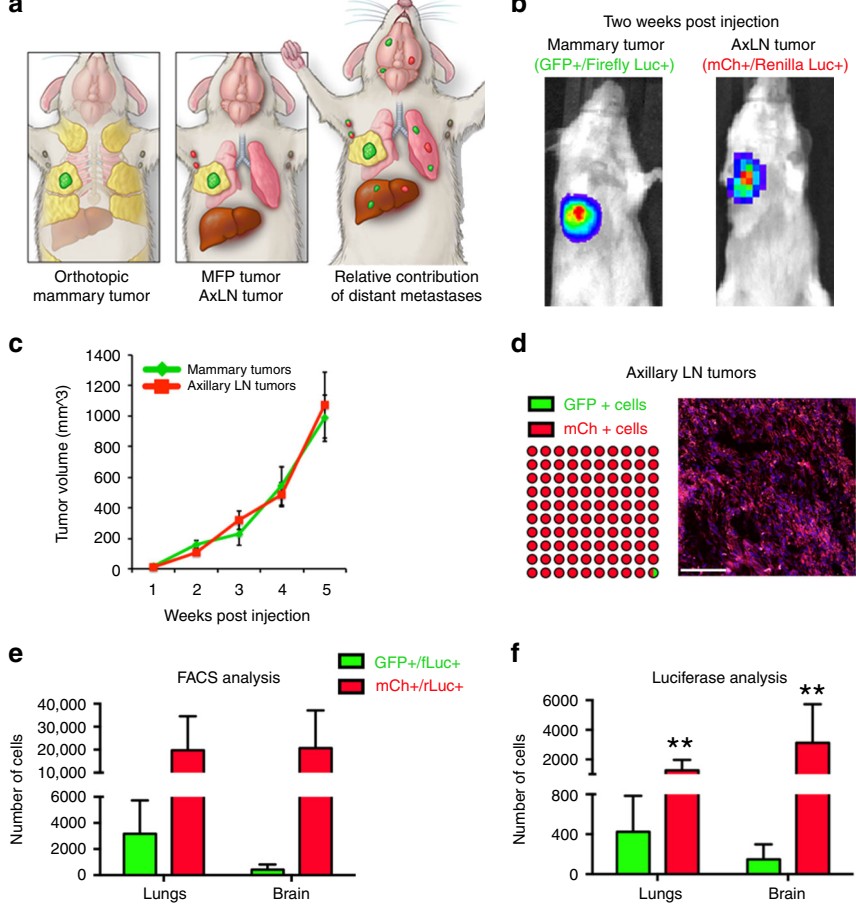

**Fig. 2** AxLN-micro-injected tumors metastasize efficiently to distant tissues. **a** Schematic depicting the injection and expected outcomes of GFP/fL- or mCh/rL-labeled 4T1 cells into the MFP or AxLNs, respectively, of the same mouse. **b** Representative IVIS imaging of GFP/fL and mCh/rL cells post injection in the same mouse. **c** Caliper measurements of tumor growth for MFP and AxLN tumors ($n = 7$ mice). **d** Fluorescence imaging of post-injected AxLNs after 6 weeks of growth show minimal GFP+ cells. **e** Flow cytometry quantification of lungs and brains to determine relative metastatic capability of tumors simultaneously established in MFPs and AxLNs. **f** Ex vivo luciferase signal observed in lungs and brains of mice simultaneously bearing GFP/fL MFP tumors and mCh/rL AxLN tumors. Statistical significance was measured by Mann–Whitney $t$ tests; **$p < 0.01$

that an epigenetic mode of gene regulation may be involved. Indeed, chromatin modifiers HDAC11 and EZH1 were significantly upregulated in the AxLN clones and were suppressed in MFP and AxLN-LuM sub-clones (Fig. 3h). We validated several of these targets by quantitative reverse transcription-PCR (Supplementary Fig. 5a). As both HDACs and histone methyl-transferases are typically involved in gene repression, we hypothesized that the increased expression of these epigenetic regulators in AxLN could be upstream of the genes that were found to be differentially down-regulated in our analyses. Using the cancer cell line encyclopedia dataset ($n = 1036$ cell lines), we found very strong inverse correlations between expression of HDAC11 and many of the candidate genes (Supplementary Fig. 5b), but not EZH1 (Supplementary Fig. 5c). Thus, we focused on further defining the role of HDAC11 in LN tumor development and subsequent spread to distant sites.

**HDAC11 is necessary for tumor growth within LNs.** HDAC11 is the most recently identified HDAC[26], and is best characterized for epigenetic inhibition of IL-10, which causes pleotropic effects on innate and adaptive immunity[27–29]. To determine whether increased HDAC11 is an artifact of micro-injection, we developed several matched pairs of sub-clones derived from MFP tumors and spontaneous AxLN metastases. As compared with corresponding MFP tumors, we found HDAC11 was significantly

increased in five out of seven matched spontaneous AxLN sub-clones; and expression levels were similar to that of micro-injected AxLN sub-clones (Fig. 3i). Using six matched primary tumors and LN metastases from the UNC RAP patient samples, we used RNA-sequencing data to evaluate HDAC11 expression. We found that some LN metastases exhibited increased HDAC11 expression compared to the matched primary tumors (Supplementary Fig. 6a, b). However, likely in part due to the small sample size and dynamic nature of HDAC11 expression, this was not statistically significant and will require further evaluation in a larger cohort of clinical samples.

Next, to determine how HDAC11 plasticity is mediated, we evaluated methylation of the HDAC11 promoter using bisulfite conversion and found that the HDAC11 promoter was less methylated in AxLN sub-clones, but had increased methylation in the MFP and AxLN-LuM sub-clones (Fig. 3j). Because decreased methylation of the HDAC11 promoter correlated with increased messenger RNA (mRNA) expression in AxLN sub-clones, these findings suggest that HDAC11 may itself be epigenetically modified in the context of the LN microenvironment; however, it is likely that other mechanisms of gene regulation are involved. To elucidate the downstream targets of HDAC11, we knocked down HDAC11 using two different short hairpins RNA (shRNAs), which resulted in significantly increased mRNA expression of several candidate genes, most notably *E2F7*, *E2F8*,

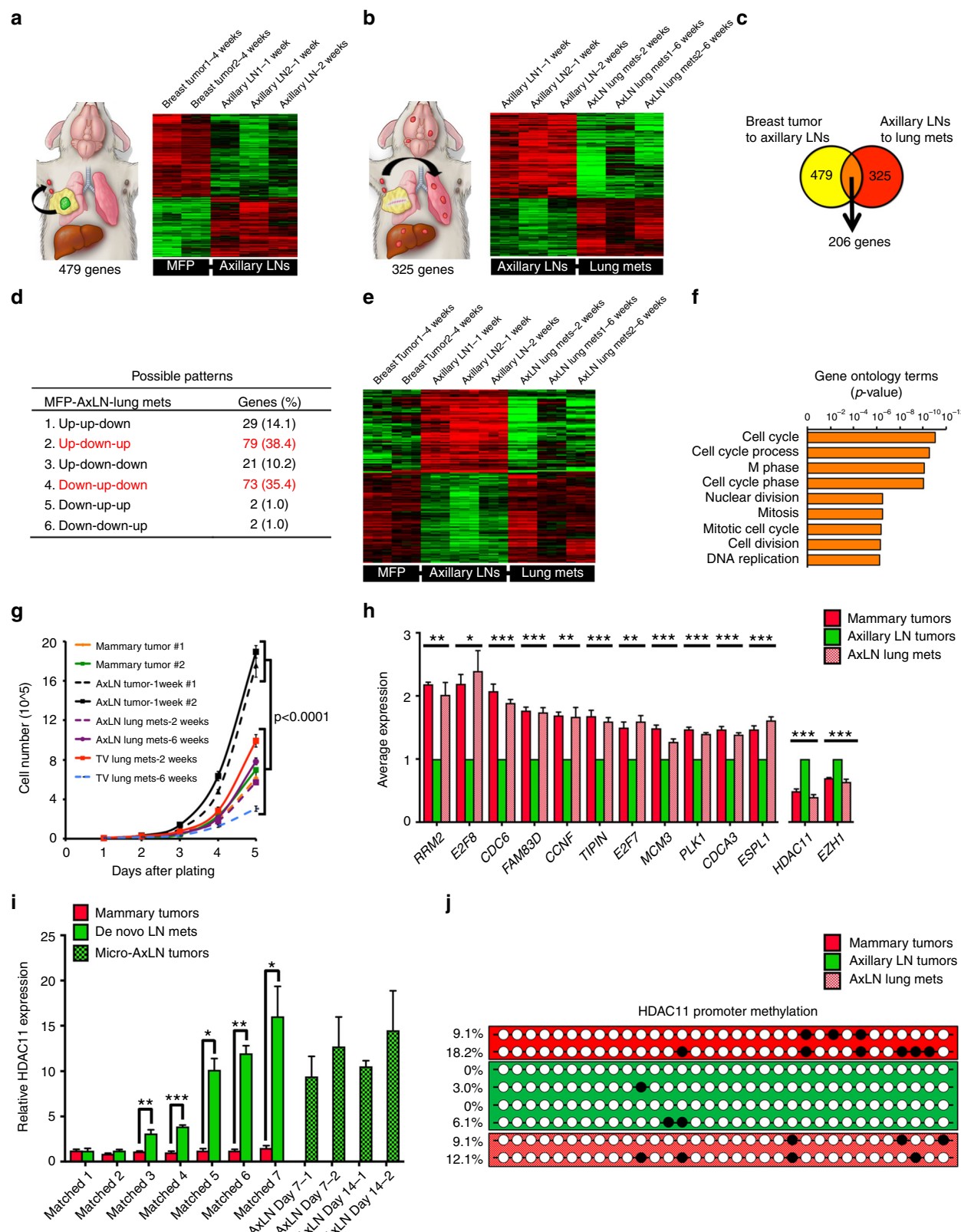

and *RRM2* (Fig. 4a). Moreover, over-expression of human HDAC11 in MDA-MB-231 breast cancer or 293 T cell lines, resulted in significant enrichment of the promoter regions of these genes by chromatin immunoprecipitation-qPCR (ChIP-qPCR) when probing for HDAC11 (Fig. 4b, Supplementary Fig. 7a). To determine whether HDAC11 was functioning as a

HDAC, we used ChIP-qPCR to compare pull-down of acetyl-H3 and acetyl-H4 at the promoters of target genes in 4T1-shCtrl and 4T1-shHDAC11 cells, which revealed significant enrichment of acetyl-H3 and -H4 at these target gene promoter regions upon HDAC11 silencing (Fig. 4c). These results were also corroborated using another triple-negative murine breast cancer cell line,

**Fig. 3** LN metastases up- or down-regulate cell cycle-associated genes in a plastic manner. **a** Pairwise gene expression array comparison between MFP- and AxLN-implanted tumors. **b** Pairwise gene expression array comparison between AxLN-implanted and AxLN-derived lung metastases. **c** Venn diagram showing shared genes between the analysis of **a**, **b**. **d** Six possible patterns of gene expression across the three experimental conditions (mammary fat pad, MFP; axillary LN, AxLN; lung metastasis derived from AxLN micro-injection, AxLN-LuM) shown in **a**, **b**. **e** Composite microarray results for the 152 genes that are up- or down-regulated in both pairwise tissue comparisons (cases 2 and 4 of **d**). Full 206 gene array shown in Supplementary Fig. 4. **f** Gene ontology analysis (biological processes) of the 152 genes displayed in **e**. **g** Cell growth assay for the ex vivo clones represented in the microarray analysis. Statistical significance was measured by unpaired $t$ tests. **h** Average expression values (RT-qPCR) for several target genes revealed by the microarray analysis. All statistical comparisons are to the axillary LN tumor samples. Statistical significance was measured by ANOVA. **i** Relative expression of HDAC11 between cell lines derived from MFP tumors and matched de novo AxLN metastasis vs. micro-injected AxLNs obtained 1 or 2 weeks post injection. **j** Bisulfite sequencing of MFP, AxLN, and AxLN-LuM sub-clones at the HDAC11 promoter CpG island. Statistical significance was measured by unpaired $t$ tests; $p$ values are indicated as *$p < 0.05$, **$p < 0.01$, ***$p < 0.001$

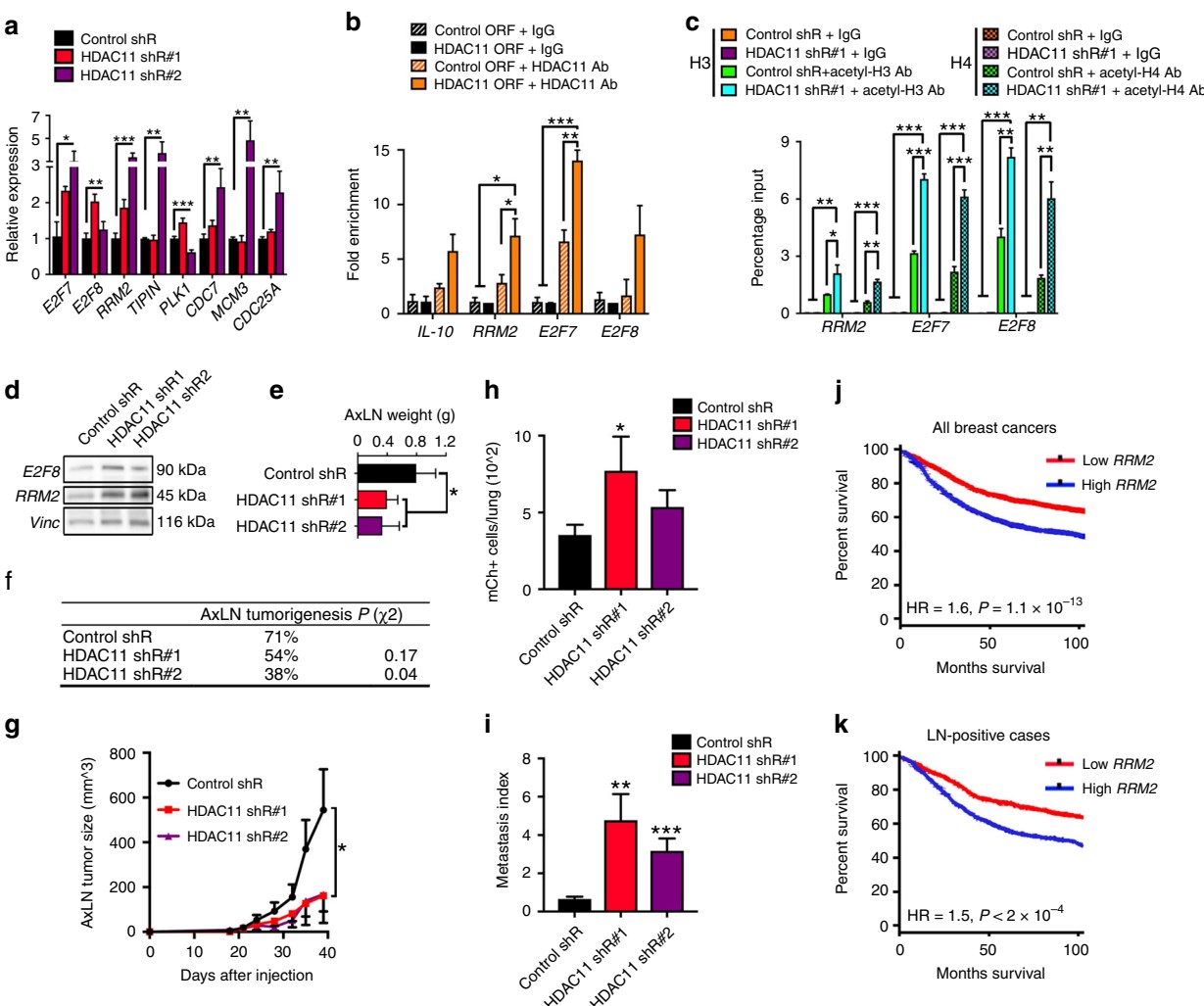

**Fig. 4** HDAC11 suppression results in reduced lymph node growth but increased metastasis. **a** Relative expression of a set of down-regulated array genes after HDAC11 suppression by shRNA. Statistical significance was measured by ANOVA in comparison to the control shR samples. **b** Levels of immunoprecipitated promoter regions for *IL-10*, *RRM2*, *E2F7*, and *E2F8* in human MDA-MB-231 cell lines expressing either control or HDAC11 ORFs when pulling down with either IgG control or HDAC11 antibodies. Statistical significance was measured by ANOVA. **c** Levels of immunoprecipitated promoter regions for *RRM2*, *E2F7*, and *E2F8* in 4T1 cell lines stably expressing either control or HDAC11 shR when pulling down with either IgG control, acetyl-H3, or acetyl-H4 antibodies. Statistical significance was measured by ANOVA. **d** Western blots for E2F8 and RRM2 in 4T1 lines expressing control or HDAC11 shRs. **e** LN tumor weights at day 35 following axillary LN micro-injection. **f** Take rates of LN-micro-injected 4T1-shHDAC11 cell lines compared to 4T1-shCtrl cells. $P$ value obtained using a $\chi^2$ contingency test. **g** LN tumor volumes throughout the duration of the experiment ($n = 13$–$14$ mice/group). Statistical significance was measured by unpaired one-sided Student's $t$ tests. **h** Lung micro-metastasis enumeration for LN-micro-injected mice by mCh + flow cytometry. **i** Lung metastasis index after normalization to LN tumor size ($n = 13$–$14$ mice/group). Statistical significance was measured by unpaired one-sided Student's $t$ tests (**h** + **i**). **j**, **k** Disease-free survival curves from the BreastMark collection comparing patients with high and low levels of *RRM2* in all available breast cancer samples ($n = 2652$ patients), as well as in LN+ cases ($n = 744$ patients). Statistical significance was measured by log-rank test. $P$ values are indicated as *$p < 0.05$, **$p < 0.01$, and ***$p < 0.001$

E0771.LMB (Supplementary Fig. 7b). Consistent with these findings, we found increased protein expression of *E2F8* and *RRM2* upon HDAC11 loss (Fig. 4d). HDAC11 knockdown also resulted in significantly reduced colony formation capability in 4T1 cells, suggesting a possible role in tumorigenesis (Supplementary Fig. 8a+b). Similarly, in E0771.LMB cells we found a dose–response of HDAC11 loss on reduced colony formation (Supplementary Fig. 8d+e). Based on these findings, we hypothesized that HDAC11 transiently increases within LNs to enable tumorigenesis in a hostile, immune-rich microenvironment. In support of this hypothesis, we observed that HDAC11 knockdown exhibited significantly reduced tumorigenicity (Fig. 4e, f) and growth kinetics (Fig. 4g) following AxLN micro-injection. Consistent with these findings, we also found that HDAC11 silencing in E0771.LMB cells resulted in significantly reduced AxLN tumorigenesis (Supplementary Fig. 8g).

**HDAC11 inhibition increased distant metastasis from LNs.** Unexpectedly, the inhibition of HDAC11 led to significantly increased LuM, especially when normalizing to the reduced AxLN tumor sizes (Fig. 4h, i). Consistent with these findings, HDAC11 silencing led to significantly increased migration in both 4T1 and E0771.LMB cells (Supplementary Fig. 8c+f). In support of decreased HDAC11 within LNs leading to increased RRM2 and distant metastasis, functioning as a "release mechanism," we found that RRM2 is highly associated with poor disease-free survival in breast cancer, including patients with LN involvement at diagnosis (Fig. 4j, k).

Because HDACis are being evaluated in numerous clinical trials in breast and other solid tumor malignancies, we next investigated whether pharmacologic inhibition of HDAC11 would yield similar effects on LN tumor growth and metastasis. Quisinostat is the most potent inhibitor of HDAC11, with sub-nanomolar potency against HDACs 1, 2, 4 and 11[30]. We found that HDAC11, but not HDAC1, 2, or 4, was upregulated in AxLN sub-clones (Fig. 5a). Treatment with quisinostat resulted in more significant cell growth inhibition in AxLN vs. MFP sub-clones with quisinostat (Supplementary Fig. 9a), suggesting that HDAC11 expression in AxLN sub-clones renders them more susceptible to HDACi. Quisinostat led to significant induction of HDAC11 target genes, yet only modest effects on genes commonly linked with an epithelial–mesenchymal transition (EMT) (Fig. 5b). Vorinostat and entinostat also led to similar dose-dependent inductions in HDAC11 target genes in a "cliff-like" pattern, whereby gene expression increased to a threshold, at which point considerable cell death was observed and gene levels dropped (Supplementary Fig. 9b, c). Protein expression of HDAC11 target genes *RRM2* and *E2F8* also increased with all 3 HDACis tested (Supplementary Fig. 9d). Based on these results, we selected sub-lethal HDACi doses that induced HDAC11 target genes for subsequent experiments (termed "sub-lethal" doses). Next, we tested the effect of sub-lethal HDACi-treatment on 4T1 growth and motility. While colony formation was considerably impeded (Supplementary Fig. 9e), transwell migration significantly increased following treatment with quisinostat, which we confirmed in E0771.LMB cells as well (Fig. 5c, d). To test the effect of quisinostat treatment on LN tumor growth and metastasis, we micro-injected AxLNs with 4T1-mCh/rL cells. After sub-palpable tumors were established as determined by luciferase signal, mice were randomly distributed and treated with either vehicle or quisinostat. Although quisinostat treatment significantly inhibited AxLN tumor growth, based on luciferase imaging and caliper measurements (Fig. 5e, f), quisinostat treatment significantly increased LuM (Fig. 5g). Notably, considering the differences in AxLN tumor size, quisinostat

increased metastasis by over 5-fold (Fig. 5h). To determine whether these results were related to the effects of quisinostat on the host vs. a cancer cell autonomous mechanism, we performed an experimental metastasis assay in which 4T1-mCh/rL cells were treated in vitro for 2 weeks with vehicle or quisinostat. No difference in micro-metastasis was seen at 24 h, whereas the quisinostat-treated cells formed significantly more LuMs by 1 week, based on mCh+ counts and hematoxylin and eosin (H&E) staining (Fig. 5i, j). Furthermore, quisinostat pre-treatment increased the average size of the lung micro-metastases (Fig. 5k), perhaps related to an increased number of pro-migratory cells extravasating and forming distant metastatic colonies. Importantly, the increased metastases were not due to a "rebound effect" as has been reported for other therapeutic agents[31], as quisinostat withdrawal resulted in colony formation rates that were similar to vehicle-treated cells (Supplementary Fig. 9f). We observed similar increases in metastatic characteristics using sub-lethal doses of vorinostat and entinostat, both of which significantly impaired cell proliferation and colony formation, but led to significantly increased migration and experimental LuM formation (Supplementary Fig. 9g–j). Although these results corroborate the phenotypes observed with genetic knockdown of HDAC11, these pharmacologic inhibitors are not HDAC11-specific and may be affecting other HDACs. Thus, it is likely that other HDACs are also involved in the phenotypes observed.

## Discussion

In nearly all cancer types, the presences of LN metastases are clinically significant and hold great prognostic power. Yet, until recently, there was no experimental evidence that LN metastases could give rise to distant metastases[18,19]. Collectively, using phylogenetic analyses of clinical samples and direct experimental models, our results show that breast cancer LN metastases can give rise to distant metastases. Our findings that the lymphatic route is highly efficient to give rise to distant metastasis carries therapeutic implications, as almost all pre-clinical models assume that cancer metastasizes by direct intravasation into and then extravasation out of blood vessels. The limited success of anti-angiogenic therapies in patients suggests that current experimental models are not fully recapitulating the metastatic process, which may in part be because lymphatic metastases are largely ignored. Recently, it was found that anti-angiogenic therapies have little effect on LN metastases, and some anti-angiogenic therapies may even promote LN metastases formation[32,33].

Herein, we have also identified a mechanism by which dissemination through LNs occurs. We demonstrate that increased expression of a chromatin modifier, HDAC11, is important for tumorigenesis and growth within the LN, but that subsequent downregulation of HDAC11 in the LN results in increased migration and egress from LNs to distant sites. We found that HDAC11 inhibits *E2F7* and *E2F8*, which are widely regarded as cell cycle suppressors[34], and this may in part explain HDAC11's role in promoting cancer cell survival within LNs (Fig. 6, top). Additionally, we found loss of HDAC11 leads to de-repression of *RRM2*. We posit that this increase in *RRM2*, which has been linked to pro-migratory and metastatic phenotypes in many cancers, including breast cancer[35], functions as a release mechanism from the LN to distant sites (Fig. 6, bottom).

In breast cancer, aberrations in histone modifications like acetylation have been shown to be important for tumor progression and prognosis, and have been proposed as a promising therapeutic target[36–38]. HDACis have been an attractive therapeutic strategy to both restore acetylation and gene expression with the potential benefit of being better tolerated than cytotoxic

**Fig. 5** Pharmacological HDAC inhibition reduces cell and tumor growth, but increases cell migration and tumor metastasis from LNs. **a** HDAC mRNA expression levels from ex vivo 4T1 clones. **b** Target gene *E2F7*, *E2F8*, and *RRM2* and EMT marker expression levels after quisinostat treatment. **c**, **d**, Transwell migration assay for 4T1 (**c**) and E0771-LMB (**d**) TNBC cells. **e** Intravital imaging of LN tumors of mice implanted with 4T1-mCh/rL cells. **f** Tumor volumes for LN-micro-injected tumor-bearing mice treated with vehicle or quisinostat (40 mg/kg; twice weekly). **g** Enumeration of lung micro-metastases in LN-micro-injected tumor-bearing mice after 6 weeks. **h** Metastasis index of vehicle- and quisinostat-treated mice after normalization to LN tumor size ($n = 10$ mice/group for **f**–**h**). **i** Enumeration of lung micro-metastases after in vitro-treated vehicle or quisinostat 4T1-mCh/rL cells were injected by the tail vein. **j** H&E staining of stitched left lungs from tail vein-injected mice. The graph shows the average number of tumors per lung for vehicle- and quisinostat-treated mice. **k** Representative H&E-stained images of lung lesions from vehicle- and quisinostat-treated mice. The graph shows individual tumor diameter measurements. Statistical significance was measured by unpaired one-sided Student's *t* tests, unless otherwise indicated; *p* values are indicated as *$p < 0.05$, **$p < 0.01$, and ***$p < 0.001$

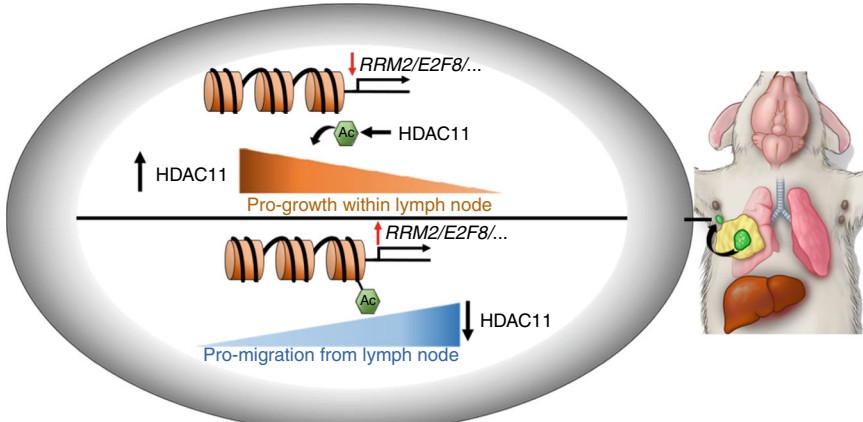

**Fig. 6** Graphical abstract of HDAC11's role in regulating lymph node metastasis. As cancer cells enter the lymph node (top), HDAC11 increases in expression, leading to decreased expression of several transcripts and increased survival within the lymph node. However, as HDAC11 decreases (bottom), the same transcripts increase in expression, leading to an increased migratory phenotype and allowing exit from the lymph node to distant organs. Thus, pharmacologic inhibition of HDAC11 may decrease cancer cell survival within lymph nodes while also increasing their migratory capabilities

chemotherapy. Epigenetic modulation has also been hypothesized to be a mechanism of resistance to endocrine therapies and cytotoxic chemotherapy[39]. Despite the promising anti-tumor effects of quisinostat and other HDACis in pre-clinical models[30], when used as monotherapy it has had limited activity thus far in patients[40]. Our results highlight the importance of evaluating candidate therapeutics in the context of LN metastasis, as well as the unique challenge of targeting plasticity in metastasis[3]. Similar to how blocking the EMT may simultaneously prevent detachment from the primary tumor and yet also promote metastatic colonization by inducing a mesenchymal-to-epithelial transition (MET), we demonstrate that the spectrum of HDAC11 expression along the metastatic cascade may have opposing effects if inhibited. Given the active investigation and development of multiple HDACis in cancer patients, our results strongly advise caution in the single-agent use of HDACi in the treatment of breast cancer and potentially other solid tumors.

## METHODS

**Cell lines and key reagents**. 4T1 cells were obtained from the ATCC and maintained in RPMI containing 10% fetal bovine serum (FBS). EO771.LMB cells were generously provided by Dr. Robin Anderson (Peter MacCallum Cancer Center). 4T1 cells were transduced with lentiviral constructs expressing either GFP/FL or mCherry/Renilla luciferase, which were obtained from Dr. Shawn Hingtgen (UNC) and were selected and maintained in puromycin (8 μg/ml). 4T1-shHDAC11 cell lines were selected and maintained in hygromycin (50 μg/ml). FLAG-HDAC11 was provided by Dr. Alejandro Villagra (George Washington University). Quisinostat, vorinostat, and entinostat were purchased from Selleckchem (S1096, S1047, and S1053, respectively). Small interfering RNAs (siRNAs) targeting HDAC11 were obtained from Sigma (siHDAC11 #1: CUAUCAAGUUC CUGUUUGAdTdT; siHDAC11 #2: GUGACAAGCGAGUAUACAUdTdT). shRNA constructs targeting HDAC11 were obtained from Genecopoeia (shHDAC11 #1: gctactcacagaacattgtca; shHDAC11 #2: ggaccactggaaataaagatt). All cells were routinely tested for mycoplasma using a Lonza MycoAlert Detection kit (LT07-418). Adult female Balb/c mice (6–8 weeks) were purchased from Taconic Farms. All animals were cared for according to guidelines set forth by the American Association for Accreditation of Laboratory Animal Care and the US Public Health Service policy on Human Care and Use of Laboratory Animals. All mouse studies were approved and supervised by the University of North Carolina at Chapel Hill Institutional Animal Care and Use Committee.

**Rapid autopsy analysis**. From the UNC Breast Cancer Rapid Autopsy Program, all clinical samples were obtained and prepared following informed consent and institutional review board approval for UNC Chapel Hill. We obtained the next-generation sequencing data for seven breast cancer patients for whom at least one primary tumor sample, one LN metastasis sample, one matched normal tissue, and multiple distant metastases samples were collected as previously described[21]. Samples with lower tumor purity were excluded from further analysis. Integrated DNA and RNA variant calling pipelines[41,42] were used for variants calling followed by a computational re-interrogation step to improve mutation calling sensitivity while keeping low false positives. Using all detected somatic mutations for each of the seven patients, we calculated a pairwise distance between sample pairs via the JSD using the R software package "phyloseq." The distance matrices reflected how many mutations of each sample diverged from the other. To determine whether the LN sample is closer to the primary or to the distant metastases in each patient, we calculated a distance ratio for each LN sample. The ratio is defined as $D_{LM}/D_{LP}$, where $D_{LM}$ is the shortest distance between the LN and any other distant metastasis and $D_{LP}$ is the distance between the LN and the primary sample in the same patient. Similarly, we used the $D_{ML}/D_{MP}$ ratio to determine whether a distant metastasis is closer to a LN sample (ratio < 1) or closer to a primary sample (ratio > 1), where $D_{ML}$ is the distance between the distant metastasis sample and the LN sample, and $D_{MP}$ is the distance between the distant metastasis sample and the primary sample in the same patient. The distance matrices were also used to construct a phylogenetic tree via the neighbor-joining algorithm (using the R software package "ape"), which produce un-rooted trees. When plotting rooted trees, we put the primary sample as its root.

In each patient, we also classified it into either "LN-met mediated" or "LN-met independent" using the same rule from ref. [10]. From each patient's rooted phylogenetic tree, if we observed a clade with only distant metastasis sample(s) and a LN sample then we concluded it as "LN-met mediated". If we do not find such cases, we concluded it as "LN-met independent."

To check the robustness of such origin classification, we performed bootstrapping of the mutation data for each patient to determine whether the origin classification was sensitive to changes in a limited number of somatic mutations. For each patient, we performed repeated random sampling of the mutation data 1000 times. In each iteration, we computed the distance matrices

and classified the patient into one of the two origin classifications as described above. We then assigned a confidence score to each patient, where the confidence score is the number of times the classification is the same as the real origin classification divided by 1000 (number of iterations) multiplied by 100.

**Microarray analyses**. Two plates of Mouse Gene 2.1 ST Affymetrix oligonucleotide microarrays were hybridized using (i) two baseline samples (i.e., 4T1-GFP-fLuc and 4T1-mCherry-rLuc), each available as a quadruplicate (four biological replicates), together with the other experimental samples, each available as a triplicate (three biological replicates). The selected experimental samples, which are shown in Fig. 2, were derived from: (1) MFP, injected in and extracted from MFP; (2) AxLN, injected in and extracted from AxLN; (3) AxLN (-derived) LuMs (AxLN-LuM), injected in AxLN and extracted from LuMs. Each sample was obtained at different timepoints; for the sake of brevity, and since we were looking for experimental evidence of time-independent gene regulation, each of these cases is called in this paragraph a "sample type." The generation (starting from the raw data) and normalization of the RNA expression values was performed using the robust multi-array algorithm[43,44]. The expression values of these two plates were scaled, according to the ratio of cumulative gene expression of each plate, and then combined into one spreadsheet. Then, the whole set of RNAs was reassessed, so that only those (i) fulfilling minimum and not conflicting annotation criteria (including predicted RefSeq, NCBI Reference Sequence Database: http://www.ncbi.nlm.nih.gov/refseq/] genes and GenBank [NIH Genetic Sequence Database: http://www.ncbi.nlm.nih.gov/genbank/] genes having an associated gene symbol) and (ii) annotated as protein-coding genes, were analyzed.

We deemed differentially expressed (vs. the baselines) genes: (i) whose average expression value for at least one sample triplet was ≥50% or ≤50% than the average in the two available baselines; (ii) ≥25th percentile in terms of range (across the samples) of the sub-matrix containing the baselines and that sample triplet, and (iii) for which each comparison between the three samples of that triplet and the baseline replicates maintained the same polarity (> or <). Afterwards, we directly compared different sample types. The comparisons performed were: (i) MFP vs. AxLN, (ii) MFP vs. AxLN-LuM, and (iii) AxLN vs. AxLN-LuM. The results of these comparisons (based on the third point above illustrated for determining genes differentially expressed) were displayed separately. Of course, no other comparisons were possible, since AxLN vs. MFP is equivalent to MFP vs. AxLN, and so on. Because of the way in which genes are selected before being displayed through heat maps as well as simple combinatorics calculations, the p values specifically associated with each gene identified by (i), (ii), and (iii) are, respectively, $1.9980e-04$, $1.9980e-04$, and $2.0568e-05$. At this point, we looked for patterns of gene expression across these three sample types, that is, for the set MFP-AxLN-AxLN-LuM, by merging these three heat maps. Every time an array RNA was shared between two of these three heat maps, it was kept in the merged heat map that displays the three sample types. This step was repeated three times, that is, as many as the possible couples of heat maps to be matched. Overall, the possible patterns of gene expression using a vocabulary containing only the words "Up" and "Down" are: (a) Up-Up-Down, (b) Down-Down-Up, (c) Up-Down-Down, (d) Down-Up-Up, (e) Up-Down-Up, and (f) Down-Up-Down. Cases a and b are considered as pattern/anti-pattern, and the same is true for cases c and d and for e and f. Due to the chosen criteria, after RNAs are collected for the heat map MFP-AxLN-AxLN-LuM following this computational protocol, it is still necessary to run an additional algorithm for associating possible genes belonging to multiple patterns to their strongest gene expression pattern. This additional algorithm assumes that when a gene reaches its highest average expression in a sample type, then that sample type receives the tag Up for that gene. Similarly, when a gene reaches its lowest average expression in a sample type, then that sample type receives the tag Down for that gene. Finally, the sample type having an intermediate average expression level is tagged as Up or Down for that gene measuring its distance from the other two sample types and assuming that if the minimum distance is from the 'Up' sample type, the sample belongs to the 'Up' sample type too, while if the minimum distance is from what is tagged as Down, the sample type is tagged as Down too. The distance $d$ used for this purpose is such that if $x_i \, \varepsilon \, X$ and $y_j \, \varepsilon \, Y$, with $1 \le i \le p$ and $1 \le j \le q$, $p$ and $q$ positive integers, then $d(X, Y) = \min_{i,j}(|x_i - y_j|)$; in our case, $p = q = 3$. Notably, in this computational framework, all the samples of a sample type are always tagged with the same tag with respect to an included gene and algorithm-based tagging is separately made for each gene. At the end of these steps, it was possible to univocally calculate how many genes belong to each of these patterns for MFP-AxLN-AxLN-LuM.

Later, the three couples of patterns/anti-patterns for the sample set MFP-AxLN-AxLN-LuM were separated and plotted in three distinct heat maps. Each hierarchical clustering was performed with respect to the matrix rows (containing the gene expression values across all samples) on log 2-trasformed, mean-subtracted data, using open source clustering software[45,46]. Each heat map was then split into its top and bottom parts, looking at the couple of genes of the heat map where a change of sign in the corresponding cdt file was present for the "divergent" sample type. We call "divergent" the sample type whose pattern tag is different from the other two (e.g., in the pattern c, Up-Down-Down, the divergent sample type is the first, MFP). In particular, the last gene having a specific sign across the divergent samples in the cdt file was considered the last gene of the "top" part and the first gene having the opposite sign was considered the first of the

"bottom" part. After this step, genes belonging to each half were separately processed using the Expression Analysis Systematic Explorer score (a *p* value derived from an adjusted Fisher's exact test) and through DAVID bioinformatics resources[47,48] for mouse genes, looking for GO terms of biological processes that were statistically significant (*p* value < 0.01).

**Breast cancer clinical outcome analysis.** To determine clinical relevance of *RRM2* expression in breast cancer subtypes, we utilized the BreastMark public microarray database of mRNA expression and clinical annotations, which has been previously described[49]. The median threshold was used to distinguish high (above median) and low (below median) expression of *RRM2*. Disease-free survival analysis was determined using the log-rank test for all patients (inclusive of luminal A, luminal B, Her2+, basal-like and normal-like subtypes) and for patients with positive LNs at the time of diagnosis.

**Real-time qPCR.** After treatment, RNA was purified from cells using the Zymo Quick RNA miniprep kit according to the protocol recommended by the manufacturer (Zymo Research, R1057). Then, complementary DNA (cDNA) was synthesized using a Bio-Rad iScript cDNA synthesis kit (1708891) for total RNA. cDNA was analyzed using SYBR Green reagent (Bio-Rad, 1525271) according to the protocol recommended by the manufacturer. PCR was done with reverse-transcribed RNA, 1 μL each of 20 μM forward and reverse primers, and 2× PowerUp SYBR Green Master Mix in a total volume of 25 μL. Data were analyzed using the ΔΔCt method, and experiments were normalized to 18S rRNA. Primer sequences included the following: Hdac11—forward (Fwd), AATGGGGCAAGG TGATCAAC and reverse (Rev), GCCACCACAAAGGACCACT; Rrm2—Fwd, ACGACCTCAACGCACAGTACG and Rev, GTAAGGGCAGGAGTCCCATGA TG; E2f7—Fwd, GATGCGTTCGTGAACTCCCTG and Rev, AGAAACTTC TGGCACAGCAGCC; E2f8—Fwd, GAGAAATCCCAGCCGAGTC and Rev, CATAAATCCGCCGACGTT; Plk1—Fwd, GTC AGAACCCATGCGGCAGCAAG and Rev, CAGGTCCACATGGTCTTCCTCTG; Hdac1: Fwd, AGTCTGTTACT ACTACGACGGG and Rev, TGAGCAGCAAATTGTGAGTCAT; Hdac2—Fwd, GCTTCGCCATCCTCGAATTACT and Rev, GTCATCACGCGATCTGTTGTAT; Hdac4—CACTGCATTTCCAGCGATCC and Rev, AAGACGGGGTGGTTGTA GGA; Cdh1: Fwd, CAGGTCTCCTCATGGCTTTGC and Rev, CTTCCGAAAAG AAGGCTGTCC; Zeb1—Fwd, GCTGGCAAGACAACGTGAAAG and Rev, GCC TCAGGATAAATGACGGC; Zeb2—Fwd, GCTACACGTTCGCCTACCG and Rev, CCTTGGGTTAGCATTTGGTGC; Twist—Fwd, GGACAAGCTGAGCAAG ATTCA and Rev, CGGAGAAGGCGTAGCTGAG; 18S—Fwd, AGAAAATCTG GCACCACACC and Rev, CTCCTTAATGTCACGCACGA; GAPDH—Fwd, CCTGACCTGCCGTCTAGAAAAACCT and Rev, CCATGAGGTCCACCA CCCTGTT. Reactions were run on a QuantStudio 6 or a StepOnePlus qPCR machine (Applied Biosystems), and cycling conditions consisted of 15 s of denaturation at 95 °C and 1 min of annealing and extension at 60 °C (40 cycles). Reactions were run in triplicate.

**Western blotting.** Cells were lysed in 0.5% Nonidet P-40 or 2% SDS. Immediately before lysis, 1x protease inhibitor cocktail (leupeptin L2884, aprotinin A1155, benzamidine B6506, trypsin inhibitor T9003, all from Sigma), 1 mM phenylmethylsulfonyl fluoride (PMSF, Sigma P7626), 1 mM NaVO$_3$ (Fisher Scientific S454-50), and 1 mM dithiothreitol (DTT, Roche 03117014001) was added to the lysis buffer. An equal amount of total protein for each sample was loaded onto 4–20% SDS-polyacrylamide gel electrophoresis (SDS-PAGE) gradient gels. After separation, proteins were transferred to nitrocellulose membranes using a Bio-Rad tank transfer apparatus. Membranes were blocked in 5% non-fat milk in phosphate-buffered saline (PBS) with 0.1% Tween-20 (PBS-T) for 1 h at room temperature, after which primary antibody was added and incubated at room temperature for 2 h or overnight at 4 °C. Anti-RRM2 was purchased from Novus Biologicals (1:1000, NBP131661), anti-E2F7 was purchased from Novus (1:1000, NBP1-80266), anti-E2F8 was purchased from Novus (1:1000, NBP152650), and anti-vinculin was purchased from Sigma (1:1000, V9131). Membranes were washed in PBS-T, after which membranes were incubated with secondary antibody at room temperature for 1 h. Membranes were washed with PBS-T, and then bands were developed using ECL reagent (Bio-Rad, Hercules, CA). Images were acquired using a Bio-Rad digital imager (Bio-Rad, Hercules, CA).

**Chromatin immunoprecipitation qPCR.** ChIP assays were conducted using the Active Motif kit (Novus Biologicals, Littleton, CO) according to the protocol recommended by the manufacturer with a few adaptations for HDAC11 immunoprecipitation (IP). HEK293T, MDA-MB-231 cells transfected with control or HDAC11-ORF, and 4T1 and EO771.LMB cells stably expressing shCtrl or shHDAC1#1 were cultured in 10-cm dishes, after which cells were incubated with 1% formalin at 37 C for 10 min. Formalin was neutralized with glycine, and then the cells were washed twice with cold PBS. Cells were scraped in cold PBS, pelleted, and resuspended in 1 mL of sodium dodecyl sulfate lysis buffer. DNA was sonicated using a Diagenode Bioruptor and then precleared with protein-A/G agarose beads. Precleared DNA was subjected to IP with 10 μg of antibody (IgG, anti-HDAC11 (both Sigma, H4539 and BioVision, 3611P were used together), anti-acetyl-H3 (Millipore 06-599), or anti-acetyl-H4 (Millipore 06-598) by rotating at

4 °C overnight. Antibody/DNA complexes were pulled down by incubation with protein-A/G beads at 4 °C for 1 h. Beads were washed, and then antibody–DNA complexes were eluted from the beads. Cross-links were reversed with 200 mM NaCl incubated at 65 °C overnight. Samples were RNase-treated and then incubated with 10 mM EDTA, 25 mM Tris-HCl (pH 6.5), and 20 mg/mL proteinase K for 1 h at 45 °C. DNA was purified using a Qiagen QIAQuick PCR purification kit, and then DNA was quantified using qPCR. Primers for ChIP-qPCR included the following: (human) E2F7—Fwd, GCTGATTGGTGGATTCTCAA and Rev, GGATCGTAGTCCCCGCTAA; E2F8—Fwd AACTTTTCCCCCAACTCTGC and Rev CCCCCGATTTGAAATTAACC; RRM2—Fwd, GCTCTCCTCACCGCATTA AC and Rev—ACAAGCGACCAGGCTTCTTA; PLK1 and Fwd—GTCCGTGT CAATCAGGTTTTC and Rev, GCTGGGAACGTTACAAAAGC; TIPIN—Fwd, CTTTCCACACTCCCACTCG and Rev, GACGTATTTCCGCGTCATCT; CDC7 —Fwd, GAAGAAACCCCACCCTCTTG and Rev, CTCCAAGAGATCCCCACC TAC; CDC25A—Fwd, CTGATTGGTGGATTCCGTTT and Rev, CACCTCTTA CCCAGGCTGTC; ESPL—Fwd, AGCCGCGGGATATTTGAAAG and Rev, ACAGG ACTTAACCGCCTGAC; MCM3—Fwd, AACAGAGAATCCCGGATGGTA and Rev, CTGAGTTCTCTGAGGTCGGACT; IL-10—Fwd, ATAAAAGGGGGACAG AGAGGTG and Rev, GCCTTCTTTTGCAAGTCTGTCT.

(Mouse) E2F7—Fwd, TTGCAAAACCCCCTTTGGTG and Rev, ACGTGAAC CCTGGTTAGCAC; E2F8—Fwd, AAGAGCCCAAACCACAATCTTA and Rev, AGTTAGGAGACCATCTCGTCCA; RRM2—Fwd, CAACTCAAATCTCCCGC GCT and Rev, TTAAAGAGCCACCCAACCGC.

**Transfection.** 4T1 cells were plated at 40% confluency in 6-well plates, after which the cells were transfected with siHDAC11 or siCtrl duplexes (60 nM final concentration; Sigma, St. Louis, MO) using RNAiMAX according to the protocol recommended by the manufacturer. The transfection was conducted for the indicated amount of time before the cells were collected and analyzed. The following siRNA constructs were used (sense strand displayed): siCtrl, UUCUCCG AACGUGUCACGUdTdT; siHDAC11-1, CUAUCAAGUUCCUGUUUGAdTdT; siHDAC11-2, GUGACAAGCGAGUAUACAUdTdT.

**Virus packaging and transduction.** Viral particles were produced by transfecting human embryonic kidney cells (HEK293T cells) with lentiviral vector, packaging plasmid (psPAX2), and envelope plasmid (pMD2G). Media were changed the next day, and 2 days later, viral supernatant was collected and filtered to remove cellular debris. For infection, 4T1 cells were plated in 6-well plates at 40% confluency. Then, viral particles were added to the cells in the presence of 8 μg/ml Polybrene. After overnight infection, the media were refreshed. Hygromycin (50 μg/ml) or puromycin (8 μg/ml) was added to the media to select for transduced cells.

**HDACi treatment.** 4T1 cells were treated in 10-cm tissue culture dishes at the indicated concentration of drug (dissolved in dimethyl sulfoxide). Media and drug were refreshed daily to ensure consistent drug exposure. For long-term treatment, cells were exposed to HDACis for at least 7 consecutive days. The sub-lethal dose for quisinostat in 4T1 cells was determined to be 10 nM, whereas vorinostat was 500 nM and entinostat was 2 μM. For the quisinostat withdrawal experiment, cells that were treated long term with 10 nM quisinostat were seeded into complete RPMI without the drug for colony formation capability or for testing subsequent passages in drug-free medium for colony formation capability.

**Haptotaxis/chemotaxis migration assays.** A total of 25,000 or 50,000 cells were added in serum-free medium to Boyden chambers (8-μm pores) that were precoated with 10 μg/ml type 1 rat tail collagen on the bottom of the inserts. RPMI medium containing 10% FBS was added to the lower chambers as the chemoattractant. Haptotaxis/chemotaxis was allowed to proceed for 18 h, after which cells were removed from the top chambers, and cells migrated to the bottom of the filter were fixed and stained using the Protocol Hema 3 staining kit (Fisher Scientific, 22122911). Membranes were mounted onto glass slides, and images were taken using a Nikon microscope. Migrated cells were enumerated using CellProfiler open source image analysis software (CellProfiler).

**Cancer cell implantation.** Adult Balb/c mice were purchased from Taconic Farms and C57Bl/6 mice were purchased from Jackson Labs. These animals were cared for according to guidelines set forth by the American Association for Accreditation of Laboratory Animal Care and the US Public Health Service policy on Human Care and Use of Laboratory Animals. All mouse studies were approved and supervised by the University of North Carolina at Chapel Hill Institutional Animal Care and Use Committee. All animals used were between 6 and 10 weeks of age at the time of injection. For all animal experiments, cells were trypsinized, washed, and resuspended in Hank's balanced salt solution (HBSS; Gibco) prior to injection.

*Mammary fat pad*: Cells were trypsinized and suspended in Matrigel at a 1:1 ratio, and 5000 cells were injected directly into the eighth MFP of anesthetized 6–10-weeks-old female Balb/c mice. Caliper measurements of subcutaneous tumor growth were taken twice weekly and the tumor volume was calculated as $L \times W^2$, where $L$ is the greatest cross-sectional length across the tumor and $W$ is the length perpendicular to $L$. Luciferase-labeled tumor progression was monitored once

weekly using an IVIS Lumina optical imaging system and Nano-Glo Luciferase Assay substrate (Promega) as per the manufacturer's instructions.

*Axillary LN*: Mice were anesthetized, depilated, and subjected to surgical implantation of 5000 (4T1) or $4 \times 10^4$ (EO771.LMB) cells in a total volume of 1 μl HBSS. Injections were performed using a dissecting microscope and a 10-μl Hamilton syringe and custom-made microtip Pasteur pipette. Caliper measurements of tumor growth were taken twice weekly, and the tumor volume was calculated as $L \times W^2$, where $L$ is the greatest cross-sectional length across the tumor and $W$ is the length perpendicular to $L$. Luciferase-labeled tumor progression was monitored once or twice weekly using an IVIS Lumina optical imaging system and RediJect Luciferase Assay substrate (Promega) as per the manufacturer's instructions. Metastasis index was determined by dividing the number of mCh cells identified by flow cytometry by the average size of the primary AxLN tumor.

*Tail vein*: Mice were injected with $5 \times 10^3$ or $1 \times 10^5$ 4T1 cells in HBSS by tail vein, after which mice were monitored daily for health. Mice were sacrificed and analyzed at the indicated time point post injection.

**Ex vivo cell line establishment and analysis**. Tumors were excised (under aseptic conditions for propagation) in a laminar flow tissue culture hood and minced using a sterile scalpel blade in digestion medium. For direct analysis, tissue was minced using scissors. Minced tissue was digested for 1 h in 0.125% collagenase II, 0.1% hyaluronidase, 15 U/ml DNase, and 2.5 U/ml dispase. Cells were then pelleted, subjected to ACK red blood cell lysis, and then pelleted and analyzed by flow cytometry or plated in 10-cm dishes containing complete RPMI medium and antibiotics as appropriate. For passaging and subsequent culture, ex vivo 4T1 sub-clones were selected with 6-thioguanine for several days until pure colonies were observed.

**Flow cytometry analysis**. Fluorescently labeled 4T1 cells were stained with Live/Dead fixable violet dye (Thermo Scientific) for 15 min, after which cells were washed and suspended in FACS buffer (1% bovine serum albumin in PBS containing 0.5 mM EDTA). Then, cells were analyzed using a Cyan (Beckman Coulter) or Attune NxT (Life Technologies) flow cytometer. FCS files were analyzed using FlowJo software (version 10; FlowJo LLC).

**H&E staining**. Tissues were fixed in 10% buffered formalin for at least 24 h, after which they were embedded in paraffin and sectioned in 4–5 μm sections. Sections were mounted onto Fisher Superfrost Plus slides and then deparaffinized and stained with H&E by the UNC Animal Histopathology Core. Stained slides were imaged using a Leica DMi8 inverted microscope at ×200 magnification. Images were scored using CellProfiler open source image analysis software (CellProfiler). Tumor diameters were measured manually by counting pixels for the largest diameter of each identified tumor.

**Colony formation assay**. After treatment, cells were trypsinized, counted, and 1000 or 5000 cells were plated in triplicate in 6-well plates containing complete RPMI medium and drug, as indicated. Cells were allowed to grow under standard conditions for at least 4 days until colonies were observed. For staining, 1 ml of crystal violet stain (0.05% crystal violet, 1% formalin, 1% methanol in PBS) was added to the cells. Cells were destained in deionized water, and images were taken using an Epson office scanner under film settings. To quantify stain, crystal violet was extracted using 1.5 ml of 1% SDS, followed by absorbance measurement at 612 nm wavelength light on a BioTek luminometer plate reader. Readings were also conducted in triplicate for each sample.

**DNA methylation detection**. Bisulfite-converted DNA was generated and purified from 4T1 sub-clones using the EZ DNA Methylation-Direct kit (Zymo Research). Bisulfite-converted DNA was then amplified by PCR using primers designed with the assistance of MethPrimer[50]: Fwd, GGTTAGAGTTTTATTTTTAGTTTTTAG; Rev, CTACAAAAAACTATACCCTCCTC.

PCR products were purified (Qiagen QIAquick PCR Purification kit) and then directly DNA sequenced (Eton Bioscience).

**Statistical analysis for in vitro and in vivo experiments**. Between 5 and 15 mice were assigned per treatment group; this sample size gave ~80% power to detect a 50% change in tumor weight with 95% confidence. Results for each group were compared using Student's *t* test (for comparisons of two groups) and analysis of variance (for multiple group comparisons). For values that were not normally distributed (as determined by the Kolmogorov–Smirnov test), the Mann–Whitney rank-sum test was used. A *P* value < 0.05 was deemed statistically significant. All other statistical tests for in vitro and in vivo experiments were performed using GraphPad Prism 7 (GraphPad Software, Inc., San Diego, CA). The multiple hypothesis testing correction of these results was made using the false discovery rate.

**Reporting summary**. Further information on research design is available in the Nature Research Reporting Summary linked to this article.

## Data availability

The microarray data that support the findings of this study have been deposited in the Gene Expression Omnibus (GEO) data bank, accession code (GSE136031).

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

## Acknowledgements

We acknowledge members of the Pecot lab for helpful discussions and feedback. We thank Dr. Alejandro Villagra (George Washington University) for providing human FLAG-HDAC11 expression plasmid. We would also like to thank the patients and their families who generously contributed to the UNC Rapid Autopsy Program. The UNC Flow Cytometry Core Facility and Lineberger Comprehensive Cancer Center Animal Histopathology and Animal Studies Cores are all supported in part by an NCI Center Core Support Grant (CA016086) to the UNC Lineberger Comprehensive Cancer Center. The UNC Flow Cytometry Core Facility is also supported in part by the North Carolina Biotech Center Institutional Support Grant 2012-IDG-1006. The UNC Breast Cancer Rapid Autopsy Program is supported by a Susan G. Komen Scholar Award (L.A.C.) and the UNC Specialized Program of Research Excellence (SPORE) P50-CA58223. E.B.H. was supported in part by a grant from the National Cancer Institute of the National Institute of Health under award number T32CA196589. C.V.P. was supported in part by the National Institutes of Health R01CA215075, a Mentored Research Scholar Grants in Applied and Clinical Research (MRSG-14-222-01-RMC) from the American Cancer Society, the Jimmy V. Foundation Scholar award, the UCRF Innovator Award, the Stuart Scott V. Foundation/Lung Cancer Initiative Award for Clinical Research, the University Cancer Research Fund, the Lung Cancer Research Foundation, the Free to Breathe Metastasis Research Award, the Susan G. Komen Career Catalyst Award, and a NCBC translational research grant.

## Author contributions

Conceptualization: C.V.P.; experimental design: P.L.L., Y.L.C., C.V.P.; data curation: A.P., Y.-H.T., J.S.P., L.A.C., A.E.D.V.S., E.B.H.; formal analysis: P.L.L., Y.L.C., S.K.G., A.P., Y.-H.T., J.S.P., C.V.P.; funding acquisition: C.V.P., L.A.C.; investigation: P.L.L., Y.L.C., S.K.G., B.C.C.; resources: C.V.P., L.A.C.; software: A.P., Y.-H.T., J.S.P.; validation: P.L.L., Y.L.C., S.K.G.; visualization: P.L.L., C.V.P; writing-original draft: P.L.L., Y.L.C., C.V.P; writing review and editing: all authors.

## Additional information

**Competing interests:** The authors declare no competing interests.

