## [Peer Review File · Nature Communications]

Reviewers' comments:

Reviewer #1 (Expertise: HDAC therapy, Remarks to the Author):

In the following manuscript entitled “Histone deacetylase 11 inhibition promotes breast cancer metastasis from lymph nodes” the authors present data that supports lymph node lesions as a source for disseminating distant metastases in breast cancer. Mechanistically, they implicate HDAC11 as a key mediator of this process, through which myriad cell cycle regulators are affected. Very intriguing is their finding that HDAC11 inhibition is efficacious against the nodal lesions, but appears to promote distant metastases. The study is robust and conclusions are supported by the data. Furthermore, this study has broad implications regarding the use of HDAC inhibition for the treatment of cancer, specifically breast cancer. The need for targeted HDAC inhibitor specificity has been an outstanding question in the scientific and clinical community. This study suggests that significant thought should be used in the design of future trials employing HDAC inhibitors with regard to disease nodal involvement. Additionally, it would be very interesting to parse findings from previous studies using HDAC inhibitors based on this relationship between nodal lesions and subsequent metastases following treatment initiation. Are patients on these trials with nodal involvement more likely to experience new lesions than those without? With this in mind, does this affect the interpretation of the therapeutic approaches, which patients should be enrolled, etc?

There are a few concerns/questions:

1. The constructed 4T1 cell lines appear to be used such that the mCherry labeled cells are always implanted in the axil lymph node and the GFP cells in the fat pad. Has this orientation been reversed to show that potential cell line divergence is not providing a selective phenotypic difference? Or 4T1 parental cells used alone to recapitulate metastatic frequency when implanted in the AxLN versus MFP?
2. The use of a single mouse cell line is always limiting. Although ER+ mouse cell lines are not available, showing some of the relationships between HDAC11 and AxLN spread with another of the triple negative lines (eg. JC, EMT6, etc.) would strengthen this as a broader mechanism in breast cancer.
3. Did the authors compare spontaneous lymph node lesions to AxLN lesions from injection with respect to HDAC11 expression? Also, comparing HDAC11 expression in the primary MFP lesion to the spontaneous node lesion in the same animal to demonstrate the relative increase when spreading?
4. The transition from relative low (MFP) to high (AxLN) HDAC11 expression remains unclear. Is this also an epigenetic switch? Is there a change in DNA methylation? Or is it more indirect?
5. Of great interest is the HDAC11 expression differences in the patient tumor samples used in the phylogenetic analysis. Demonstrating this possibly by RNA scope, using FFPE blocks if available, would tie this mechanistic work back to the clinic.

Regards,

Scott Thomas, PhD

Helen Diller Family Comprehensive Cancer Center

University of California, San Francisco

Reviewer #2 (Expertise: Metastasis in vivo modeling, Remarks to the Author):

In this manuscript Leslie et al suggest that HDAC11 controls breast cancer metastasis from lymph nodes. The authors develop an experimental model of metastasis from the lymph node that they use to compare growth as well as metastatic properties. Subsequently they engage in a path to identify genes and functions that may contribute to trigger metastasis from the lymph nodes. This is an interesting topic of much clinical debate with implications still uncertain today. Overall, the manuscript covers an interesting and relevant subject, is experimentally well-executed although it suffers from some conceptual and technical inconsistencies that must be address to build on solid ground.

Conceptual limitation:

Interestingly, through a genetic analysis of a limited subset of patient samples, the authors come to the interesting observation that in some patients, lymph node and distant metastasis are phylogenetically related in what suggest to be co-occurrent events in time and space. However, the distant metastasis does not represent a subclone of the lymph node met. This is interesting and relevant, yet clashes with the observation in the experimental model that lymph node implanted cells produce more metastasis than MFP. In addition, this is enhanced in the absence of HDAC11. This are puzzling observations, conceptually counterintuitive and question to what extend the experimental model recaps what is observed in the patients. Please address

Major points:

1- The major difference reported in the genes and functions specifically enriched in AxLN clones are involved in cell cycle progression (Fig 3f). This is not surprising and well established. Passaging cells in vivo may select for highly proliferative clones and is well established that

proliferation and tumor size are more prone to metastasize. Thus, I am concern on whether the model is relevant compared with what is observed in the patients. To what extend this is an enrichment in proliferation markers per se or a loss of differentiation attributes.

2- It is unclear what is the link and basis for the selection and focus of HDAC11, why was it prioritize as opposed to others. Please elaborate. Next, the data based on HDAC11 depleted cells suggested uncoupling between proliferation and metastasis, yet HDAC11 was selected on the basis of controlling genes supporting proliferation. Again, unclear. Please streamline the message and provide a clear reasoning for the work.

3- All data provided except one plot are based on PCR or similar mRNA-based approaches. Whereas functions are performed by proteins. Please provide western blot quantification for HDAC11 downregulation. What about RRM2 or EZH2? Similarly, how many times has figure 4a been performed. The differences are marginal to be responsible for such significant effects. Please clarify.

4- The authors show ChIP data based on HDAC11 overexpression in 293 cells (Fig 4c). This is interesting but limited. Please provide Histone Acetylation marks occupancy in such promoters in the absence or presence of HDAC11. Interacting at the TSS of a handset of genes is interesting but confirmation of HDAC11 function is needed beyond the use of a chemical inhibitor to confirm that is via chromatin remodeling and not direct deacetylation of transcription like proteins that HDAC11 is mediating the effect.

5- All figures in the paper are performed with one cell line except on panel performed on E0771 (Fig 5d). This is too limited to a particular experimental question (transwell migration) and falls short as a generalization of the findings. The authors must recap some of the critical in vivo experiments with this alternative cell system.

6- Overall, the mechanistic explanation on how HDAC11 supports metastasis is scarce. Migration is relevant but may explain an increase in the number of new foci generated (as a direct readout of more cells disseminated), yet the authors observed an increase in the size of the lesions (Fig 5 j/k) which can unlikely be explain by increase migratory properties. This needs to be addressed.

Minor

In the intro (line 48) "knowledge gap" is repeated twice in the same sentence.

Reviewer #3 (Expertise: Breast cancer metastasis, Remarks to the Author):

This manuscript describes the investigation of the role of HDAC11 in breast cancer metastasis. Using a microinjection technique to introduce tumor cells into axillary lymph nodes the investigators compare the ability of lymph node injected tumors to disseminate with that of cells injected into the mammary fat pad. The investigators find a greater number of disseminated tumor cells in distant organs in the lymph node injected animals, suggesting a more permissive environment for dissemination than the mammary fat pad. The investigators subsequently perform expression analysis and identify HDAC11 as a possible mediator of the lymph node tumor growth potential, which they evaluate using pharmacological approaches. The authors conclude from their work that HDAC11 promotes breast cancer growth and suggest that use of HDAC inhibitors may be counter-indicated for this malignancy.

Comments:

A major concern regarding the analysis of the SNV data revolves around sequencing of the primary tumor. Since tumors evolve into distinct subclones, distant measurements between primary tumor and distant metastases may be affected by which part of the primary tumor is analyzed. If, for example, a fraction of the primary that did not contain the subclone that disseminated was sequenced, distant and lymphatic metastases may appear to be more closely related to each other than to the primary, giving the appearance of a lymph node-to-distant site seeding. Similarly, since subclones can arise or be extinguished in the primary tumor evolution, it is also possible that the subclone that disseminated at an early time point might be extinguished later, again making it appear that lymph node and distant metastases are more closely related than to the primary tumor. Thus, it is difficult to interpret the possibility of the lymph node-to-distant site spreading unambiguously.

The 4T1 cell line assignment as a TNBC surrogate is somewhat controversial. While the cell line does not express either ER or PR, transcriptionally it resembles luminal breast cancer based on a PAM50 equivalent gene expression signature (PMID:28430642).

Extended Data Figure 2: The facs plots as I understand the manuscript indicate the time course of initial dissemination of labelled cells to the organs, not necessarily metastasis, which is the formation of a multicellular lesion at the secondary site. This is an important distinction in my opinion, since the lungs have a very significant increase in tumor cells which would be consistent with formation of metastases, while the brain and liver show dissemination, but not the massive increase that one would expect from the formation of a clinically relevant proliferative lesion. Without histology to demonstrate the presence or absence of multicellular lesions, I do not feel that either the brain or liver should be considered colonized, based on this data. In addition, the comparing the number of cells in the lung from an implanted tumor in the lymph node and a bolus injection of cells into the tail vein is not an appropriate comparison, assuming the legend of panel f of this figure is correct.

Figure 3H, unpaired T-tests are not the correct statistical test here. Anova with correction for multiple testing of the mammary tumor and lung met samples against the LN tumors would be the more appropriate test. This data is also counter-intuitive based on figure 3g since suppression of RRM2 is associated with cell proliferation inhibition (PMID: 29749541) and E2F8 promotes proliferation (PMID: 26992224). How do the authors reconcile this discrepancy?

Figure 4b, again, ANOVA is the more appropriate statistical test. How do the authors reconcile the increase in E2F8 expression seen here with the decreased RRM2 and E2F8 expression seen with higher HDAC11 expression shown in figure 3h? CHIP-seq of HDAC11 in HEK-293T cells, which are kidney cells, may not represent relevant chromatin binding sites present in breast epithelial cells. Performing this in MBA-231 or some other breast cancer cell line would be more appropriate.

UNC
SCHOOL OF MEDICINE

Chad V. Pecot, M.D.
Assistant Professor

UNC Lineberger Comprehensive Cancer Center
Thoracic Medical Oncology
450 West Drive, Office 32.048
Chapel Hill, NC 27599
Office: 919-966-4779, Cell: 615-305-3955
pecot@email.unc.edu

June 21st, 2019

Dear Reviewers:

On behalf of my co-authors, I thank the referees for the review of our manuscript entitled, "Histone deacetylase 11 inhibition promotes breast cancer metastasis from lymph nodes" (NCOMMS-18-33493-T) to be considered for publication in *Nature Communications*. We are deeply appreciative of the reviewers' comments and believe the subsequent experiments we have performed substantially improved the scope and impact of our manuscript. The purpose of this letter is to highlight all of these changes to the manuscript, and I have outlined these below:

Reviewer #1 Comments:

Mechanistically, they implicate HDAC11 as a key mediator of this process, through which myriad cell cycle regulators are affected. Very intriguing is their finding that HDAC11 inhibition is efficacious against the nodal lesions, but appears to promote distant metastases. The study is robust and conclusions are supported by the data.

We thank the reviewer for these supportive comments.

1) The constructed 4T1 cell lines appear to be used such that the mCherry labeled cells are always implanted in the axil lymph node and the GFP cells in the fat pad. Has this orientation been reversed to show that potential cell line divergence is not providing a selective phenotypic difference? Or 4T1 parental cells used alone to recapitulate metastatic frequency when implanted in the AxLN versus MFP?

We thank the reviewer for these comments, and the subsequent experiments have further bolstered our original findings. We have performed two *in vivo* experiments to address the reviewer's concern that there may be intrinsic phenotypic differences between 4T1-GFP and 4T1-mCherry labeled cells that may have accounted for the different metastatic efficiencies. In the first experiment, we switched the reporter orientations, as suggested by the reviewer, and compared metastatic efficiency of 4T1-GFP/fLuc cells micro-injected into AxLN with 4T1-mCh/rLuc cells injected into MFP. We found that 4T1-GFP/fLuc cells injected into AxLN were significantly more capable of establishing distant metastases in the lung compared to 4T1-mCh/rLuc cells injected in MFP (Extended Data Fig 3a). In the second experiment, we directly compared frequencies of distant metastases of 4T1-mCh/rLuc cells injected into either MFP or AxLN. To further delineate the contribution of hematogenous versus lymphatic dissemination from MFP, we also injected a group of mice with cells into the MFP after AxLNs were surgically removed. The metastasis index of 4T1-mCh/rLuc cells injected into AxLN was significantly increased compared to metastasis from MFP with AxLN removed (Extended Data Fig 3b). Taken together, these two experiments provide additional support for our findings that the lymphatic route is an

efficient route of spread in establishing distant metastases, and potentially more efficient than the hematogenous route.

2) The use of a single mouse cell line is always limiting. Although ER+ mouse cell lines are not available, showing some of the relationships between HDAC11 and AxLN spread with another of the triple negative lines (eg. JC, EMT6, etc.) would strengthen this as a broader mechanism in breast cancer.

We agree with the reviewer that the use of a single mouse model is limiting, however this also represents a shortcoming in the field. Unfortunately, there are very few immune-competent mouse models of breast cancer that can spontaneously metastasize to loco-regional lymph nodes and distant sites. For this reason, we obtained E0771.LMB due to it being triple-negative and its ability to spontaneously metastasize in immune-competent mice (Johnstone et al, *Disease Models & Mechanisms*, 2015). We have accordingly repeated several key experiments and we found a dose-response relationship between HDAC11 loss and decreased colony formation and increased migration *in vitro* (Extended Data Figure 8d-f), which support our previous findings. When replicating the *in vivo* experiment comparing micro-injected E0771.LMB-shCtrl and -shHDAC11 cells into AxLNs (n=13-14 mice/group), we did find a significant reduction in AxLN tumorigenicity in the HDAC11 shR groups (Extended Data Fig 8g); however, despite using large numbers of mice, due to low take rates in the AxLN across all groups we were unable to assess differences in distant metastases.

3) Did the authors compare spontaneous lymph node lesions to AxLN lesions from injection with respect to HDAC11 expression? Also, comparing HDAC11 expression in the primary MFP lesion to the spontaneous node lesion in the same animal to demonstrate the relative increase when spreading?

We thank the reviewer for this question as it has helped further bolster the validity of our micro-surgical model. To address this concern, we injected 4T1-mCh/rLuc cells into the MFP, and upon primary tumors reaching ~12-15 mm in diameter, we isolated sub-clones from matched MFP tumors and spontaneous *de novo* LN metastases. We then evaluated HDAC11 expression by qPCR. Importantly, we observed significantly increased HDAC11 expression in spontaneous LN metastases compared to the primary MFP tumor in the majority (5 out of 7) of matched pairs. Also, the relative fold-change in expression was similar to the degree observed in our micro-surgically injected AxLN sub-clones (revised Fig 3i).

4) The transition from relative low (MFP) to high (AxLN) HDAC11 expression remains unclear. Is this also an epigenetic switch? Is there a change in DNA methylation? Or is it more indirect?

We appreciate this interesting reviewer question. To determine whether the apparent plastic changes in HDAC11 expression is epigenetically regulated, we performed bisulfite conversion to determine the DNA promoter methylation status of HDAC11 in our MFP, AxLN, and AxLN-LuM sub-clones. Interestingly, we found that the HDAC11 promoter of AxLN sub-clones were hypo-methylated compared to MFP and AxLN-LuM subclones (revised Fig 3j). This interesting finding demonstrates that chromatin modifiers, like HDAC11, can themselves be epigenetically regulated in the context of LN metastases.

5) Of great interest is the HDAC11 expression differences in the patient tumor samples used in

the phylogenetic analysis. Demonstrating this possibly by RNAscope, using FFPE blocks if available, would tie this mechanistic work back to the clinic

In order to address this question, we were able to obtain HDAC11 expression levels by RNA-Seq for 6 out of the 7 patients from which there were matched primary and LN metastases. Although we observed a consistent pattern of increased HDAC11 expression in the LN metastasis, this was not statistically significant (Extended Data Figure 6). This was not surprising not only due to the small sample size, but also because our data suggest that increased HDAC11 expression in LN metastasis is both dynamically altered (plastic) and may be difficult to capture in patient samples. However, we do believe these findings are supportive of our preclinical findings and will need to be further investigated on larger cohorts of matched primary and LN metastasis samples once available.

Reviewer #2 Comments:

This is an interesting topic of much clinical debate with implications still uncertain today. Overall, the manuscript covers an interesting and relevant subject, is experimentally well-executed although it suffers from some conceptual and technical inconsistencies that must be addressed to build on solid ground.

We thank the reviewer for the positive comments.

1) Conceptual limitation: Interestingly, through a genetic analysis of a limited subset of patient samples, the authors come to the interesting observation that in some patients, lymph node and distant metastasis are phylogenetically related in what suggest to be co-occurrent events in time and space. However, the distant metastasis does not represent a subclone of the lymph node met. This is interesting and relevant, yet clashes with the observation in the experimental model that lymph node implanted cells produce more metastasis than MFP. In addition, this is enhanced in the absence of HDAC11. This are puzzling observations, conceptually counterintuitive and question to what extent the experimental model recaps what is observed in the patients. Please address.

We appreciate the reviewer's comment and now realize the way we originally presented these data was unclear. We have now revised the results and discussion section to better articulate these findings. The purpose of these analyses was to determine whether distant metastases were more phylogenetically similar to LN metastases in clinical samples. Indeed, our results showed that in most patients (5 out of 7), distant metastases had evidence of being more similar to LN metastases than to the primary tumor, suggesting that distant metastases could in fact represent a sub-clone of the lymph node metastasis as described by the reviewer. These findings indeed support our experimental models that LNs can serve as an important hub for distant metastases to form. We provide these data to give a clinical context for our experimental model, however only the latter is able to provide more definitive evidence of LN metastasis giving rise to distant metastases.

2) The major difference reported in the genes and functions specifically enriched in AxLN clones are involved in cell cycle progression (Fig 3f). This is not surprising and well established. Passaging cells in vivo may select for highly proliferative clones and is well established that proliferation and tumor size are more prone to metastasize. Thus, I am concerned on whether the model is relevant compared with what is observed in the patients. To what extent this is an enrichment in proliferation markers per se or a loss of differentiation attributes.

We appreciate the reviewer's concerns. However, with regards to the concern that the passaging of the cells *in vivo* could select for highly proliferative clones, all of the groups (MFP, AxLN and AxLN-LuM) were passaged *in vivo* and expanded *ex vivo* prior to microarray analysis. Therefore, even if these conditions selected for a higher proliferative phenotype, the AxLN sub-clones were still far more proliferative than the parental lines and MFP and AxLN-LuM sub-clones. We agree that while the finding of a proliferation signature is common and may not be particularly surprising, these findings are consistent with what has been observed in breast cancer patients. As described by Whitfield et al "Common Markers of Proliferation", *Nature Reviews Cancer*, 2006 (PMID 16491069), expression of genes involved in cell growth and proliferation have been found to be increased in almost all microarrays and is nearly always associated with a poor prognosis. Based on this context, we believe our finding that AxLN sub-clones having distinctly increased proliferative signatures are very consistent with these prior observations and support them functioning as a 'hub' for metastasis dissemination.

3) It is unclear what is the link and basis for the selection and focus of HDAC11, why was it prioritize as opposed to others. Please elaborate.

We thank the reviewer for this comment, and now realize that the rationale behind the selection of HDAC11 was previously unclear. We have now clarified in the results why we selected HDAC11 over other candidates (e.g. EZH1). In summary, when we divided differentially expressed genes into possible patterns of expression based on subgroup (MFP, AxLN, and AxLN-LuM) and direction of expression, the two predominant patterns were Up-Down-Up and Down-Up-Down (Figure 3d). We hypothesized that epigenetic regulators would most likely be involved in such gene expression plasticity. We focused on HDAC11 because it was one of the most highly-upregulated genes in AxLN sub-clones, it is a chromatin modifier, and its biological roles in cancer have not been studied. Also, in contrast to the other epigenetic candidate on our microarray (EZH1), only HDAC11 expression was strongly inversely correlated with many of the genes that were found in our microarray analyses (Extended Data Fig 5).

4) Next, the data based on HDAC11 depleted cells suggested uncoupling between proliferation and metastasis, yet HDAC11 was selected on the basis of controlling genes supporting proliferation. Again, unclear. Please streamline the message and provide a clear reasoning for the work.

We agree with this reviewer comment, and we have now improved the clarity of our findings in the discussion section and also added a graphical abstract (new Figure 6) to streamline how HDAC11 mechanistically contributes to distant metastasis from LNs. As illustrated in the graphical abstract, increased expression of HDAC11 is important for growth within LNs (top); however, both genetic and pharmacologic inhibition of HDAC11 within LN metastasis leads to increased migration and subsequently metastasis to distant sites (bottom). Our evidence suggests that HDAC11 up-regulation is necessary for growth and proliferation with LNs (Fig 4e-g, Extended Data Fig 8b+e+g), but then down-regulation of HDAC11 within LNs is important for migration away from LNs to distant metastatic sites (Fig 4h+i, Extended Data Fig 8f).

5) All data provided except one plot are based on PCR or similar mRNA-based approaches. Whereas functions are performed by proteins. Please provide western blot quantification for HDAC11 downregulation. What about RRM2 or EZH2? Similarly, how many times has figure 4a

been performed. The differences are marginal to be responsible for such significant effects. Please clarify

We agree with this comment, and we have now added data showing increased protein expression of HDAC11's downstream targets, RRM2 and E2F8, in 4T1-shHDAC11 cells (revised Fig 4d) as well as following treatment with several HDAC inhibitors (Extended Data Fig 9d). However, we now share the reviewer's concern that the protein levels of HDAC11 previously show in Fig 4a did not adequately reflect the degree of changes we found by qPCR in our HDAC11 shRNA sub-clones, nor in the observed degree of phenotype. As shown in (a, top) in the figure here (right), degrees of HDAC11 loss are not consistently reflective of that seen by qPCR (a, bottom). Of note, we used several primer pairs for HDAC11 qPCR, all of which gave consistent results. To further investigate this concern, we generated several sub-clones of HDAC11 CRISPR knock-out lines (b) which we sequence verified for loss of HDAC11. Again, we found inconsistent protein levels (b, top) of HDAC11 in comparison to qPCR (b, bottom). Because the antibody we previously used is polyclonal (from Sigma), we then tried two other vendors (Abcam and Santa Cruz, the latter of which is a monoclonal antibody), however, neither of these antibodies performed well. Along these lines, it is noteworthy that we had to use 2 different HDAC11 antibodies for our CHIP protocol, which is based on the protocol developed by Villagra et al, *Nature Immunology*, 2009. To add rigor to our CHIP experimental approach, we included use of a human HDAC11 ORF (new Fig. 4b and Extended Data Fig. 7a).

In summary, we found that our ability to cleanly detect only HDAC11 by western blot is limited due to lack of quality antibodies. Because HDAC11 is the most recently identified of all HDACs and it has lots of sequence homology with HDACs 1-3, we speculate that the Sigma antibody is likely detecting other HDACs beyond HDAC11. Due to these concerns, we believe it is most prudent to remove the previously presented data in Fig 4a (right), which demonstrated increased HDAC11 protein expression in AxLN sub-clones as compared with MFP and AxLN-LuM sub-clones.

HDAC11 Ab does not correspond consistently with degree of silencing detected by qPCR in both shRNA and CRISPR knock-out sub-clones.

Previously shown Figure 4a showing HDAC11 expression in several 4T1 sub-clones from mammary fat pad (MFP), axillary LNs (AxLNs) and lung metastasis derived from AxLN micro-injection (AxLN LuM).

6) The authors show ChIP data based on HDAC11 overexpression in 293 cells (Fig 4c). This is interesting but limited. Please provide Histone Acetylation marks occupancy in such promoters in the absence or presence of HDAC11. Interacting at the TSS of a handset of genes is interesting but confirmation of HDAC11 function is needed beyond the use of a chemical inhibitor to confirm that is via chromatin remodeling and not direct deacetylation of transcription like proteins that HDAC11 is mediating the effect.

We thank the reviewer for this comment, and the results obtained based on this suggestion greatly strengthened the manuscript by showing that HDAC11 is functioning at these downstream target genes. We have addressed this comment in two ways:

1. We agree that HDAC11 overexpression in 293 cells may not be the best representation of what is occurring in breast cancer cells. We therefore have over-expressed HDAC11 in MDA-MB-231 cells, a widely used triple-negative human breast cancer cell line (we have a human HDAC11-ORF plasmid, which is why this was not done in 4T1 cells). We have repeated the ChIP experiment using 2 anti-HDAC11 antibodies on 231-control ORF and 231-HDAC11 ORF cells and found that HDAC11 is present in the promoters of several target genes (Fig 4b).
2. To confirm HDAC11 function, using both 4T1 and E0771.LMB cell lines with and without HDAC11 expression, we have performed ChIP-qPCR using anti-acetyl-H3 and anti-acetyl-H4 antibodies and compared acetylation of H3 and H4 at the promoters of target genes RRM2, E2F7, and E2F8 (Fig 4c and Extended Data Fig 7b). In both cell lines, knock-down of HDAC11 expression resulted in increased acetylation of H3 and H4 at key target genes.

7) All figures in the paper are performed with one cell line except on panel performed on E0771 (Fig 5d). This is too limited to a particular experimental question (transwell migration) and falls short as a generalization of the findings. The authors must recap some of the critical *in vivo* experiments with this alternative cell system.

Similar to Reviewer #1, Comment #2, we agree that the single model is limiting. Previously the use of E0771.LMB as an alternative model was used for *in vitro* quisinostat experiments. As described above, compared with E0771.LMB-Control shR, E0771.LMB-shHDAC11 cell lines were found to replicate the key *in vitro* phenotypes that we observed in 4T1-shHDAC11 cells. Specifically, we found a dose-response relationship between HDAC11 loss and decreased colony formation and increased migration *in vitro* (Extended Data Figure 8d-f). When replicating the *in vivo* experiment comparing micro-injected E0771.LMB-shCtrl and -shHDAC11 cells into AxLNs (n=13-14 mice/group), we found a significant reduction in AxLN tumorigenicity in the HDAC11 shR groups (Extended Data Fig 8g). However, due to low take-rates in the AxLN we were unable to assess differences in distant metastases before mice had to be sacrificed due to significant tumor burden in the LN tumors. As mentioned in response to Reviewer #1, Comment #2, there are very limited breast cancer cell lines that grow in immune-competent mice and spontaneously metastasize to both lymph nodes and distant sites. This is a current limitation in the field, however one we and others are hoping to improve upon.

8) Overall, the mechanistic explanation on how HDAC11 supports metastasis is scarce. Migration is relevant but may explain an increase in the number of new foci generated (as a direct readout of more cells disseminated), yet the authors observed an increase in the size of the lesions (Fig 5 j/k) which can unlikely be explain by increase migratory properties. This needs to be addressed.

We appreciate the review's comment, and to improve clarity we have made several efforts to elaborate on these specific findings in the discussion section and through a new graphical abstract (Fig 6). The focus of the work described in this paper is on how HDAC11 contributes to metastasis from lymph nodes. Our mechanistic explanation is that increased HDAC11 expression in LN is important for cancer cell survival and proliferation, but that subsequent down-regulation of HDAC11 in the LN leads to migration and metastasis from LN to distant sites. As pointed out by the reviewer, this mechanism is supported by the increase in the number of new foci generated (Fig 5j). Our data support

the hypothesis that there are factors specific to the LN microenvironment that increase HDAC11 expression resulting in increased growth in LN. Similarly, there are likely factors that are specific to distant sites that support growth of cancer cells in those organs that are not addressed in this manuscript and can explain the increased growth in the lung that is observed. However, there are several ways that increased migration from LN could lead to increases in the size of lung metastases as observed: 1) HDAC11 inhibition may lead to increased migration at an earlier time point compared to controls, in which case distant metastases have increased time for colonization and growth within distant sites, and 2) migration occurs both as single cell and as clusters as has previously been demonstrated (Aceto et al, *Cell*, 2014).

9) Minor: In the intro (line 48) “knowledge gap” is repeated twice in the same sentence.

We thank the reviewer for catching this and it has been corrected in the manuscript.

Reviewer #3 Comments:

1) A major concern regarding the analysis of the SNV data revolves around sequencing of the primary tumor. Since tumors evolve into distinct subclones, distant measurements between primary tumor and distant metastases may be affected by which part of the primary tumor is analyzed. If, for example, a fraction of the primary that did not contain the subclone that disseminated was sequenced, distant and lymphatic metastases may appear to be more closely related to each other than to the primary, giving the appearance of a lymph node-to-distant site seeding. Similarly, since subclones can arise or be extinguished in the primary tumor evolution, it is also possible that the subclone that disseminated at an early time point might be extinguished later, again making it appear that lymph node and distant metastases are more closely related than to the primary tumor. Thus, it is difficult to interpret the possibility of the lymph node-to-distant site spreading unambiguously.

We agree with all of the astute points raised by the reviewer. Analysis of patient samples is inherently limited due to temporal and spatial heterogeneity, not only at the time of tissue collection but also across the evolution of disease within each patient. We provide this clinical data not to show unambiguously that distant metastases come from LN metastases, but to provide indirect correlative evidence that it can occur. It is for this reason that we then turned to use of experimental models to bolster our findings of the phylogenetic analyses and to investigate mechanisms of LN metastasis. We have included the caveats mentioned by the reviewer in our discussion section to discuss these possible shortcomings inherent in analyzing molecular data from clinical samples.

2) The 4T1 cell line assignment as a TNBC surrogate is somewhat controversial. While the cell line does not express either ER or PR, transcriptionally it resembles luminal breast cancer based on a PAM50 equivalent gene expression signature (PMID:28430642).

We thank the reviewer for providing this reference. We have clarified in the manuscript that the 4T1 cell line is pathologically a triple-negative subtype but is defined by a luminal molecular subtype. However, we believe this model is still relevant for several reasons:

- 1) Among patients with triple-negative breast cancer, all of the molecular subtypes are represented (PMID 26253814), making this still a clinically-relevant model.
- 2) As pointed out in the 28430642 reference, the 4T1 model is one of the few syngeneic, immunocompetent breast cancer models. Due to the scarcity of models, 4T1 is most widely used, specifically in recent papers that have advanced what is known about LN metastasis (see Brown et al and Pereira et al, *Science*, 2018 sister papers). Of

note, we now also include data with the E0771.LMB model, which was also characterized in this reference provided by the reviewer and was confirmed to be triple-negative with a claudin-low PAM50 signature.

3) Extended Data Figure 2: The facts plots as I understand the manuscript indicate the time course of initial dissemination of labelled cells to the organs, not necessarily metastasis, which is the formation of a multicellular lesion at the secondary site. This is an important distinction in my opinion, since the lungs have a very significant increase in tumor cells which would be consistent with formation of metastases, while the brain and liver show dissemination, but not the massive increase that one would expect from the formation of a clinically relevant proliferative lesion. Without histology to demonstrate the presence or absence of multicellular lesions, I do not feel that either the brain or liver should be considered colonized, based on this data.

We appreciate the reviewer's comment and agree that inclusion of the brain and liver metastases data should not be included since we did not verify metastatic colonization at these sites like we did for the lung metastases. We have removed the brain and liver data from Extended Data Fig 2.

4) In addition, the comparing the number of cells in the lung from an implanted tumor in the lymph node and a bolus injection of cells into the tail vein is not an appropriate comparison, assuming the legend of panel f of this figure is correct.

We now realize our figure legend was unclear, which we have now clarified. The same number of cancer cells (5×10^3 suspended in HBSS) were either micro-injected into the AxLN (lymphatic) or directly into the tail vein (hematogenous), and then we compared the number of cells in the lung at the time-points (24hrs, 72hrs, 1, 2 and 6 wks) between the two routes (Extended Data Fig 2b-f). Anatomically the AxLN drains into the same lung venous network as the tail vein, either via the efferent lymphatic duct draining into the subclavian vein, or through draining AxLN vessels. Because the entire 5×10^3 bolus of cells via the tail vein route reaches the lungs, instead of the AxLN where many cells likely die upon injection, this would have theoretically biased the results towards more lung metastases via the hematogenous route. Thus, we believe our findings that lung metastases formed at a higher number, and frequency, from the lymphatic route are an appropriate comparison and also quite meaningful.

5) Figure 3H, unpaired T-tests are not the correct statistical test here. Anova with correction for multiple testing of the mammary tumor and lung met samples against the LN tumors would be the more appropriate test.

We have re-evaluated the statistics for this figure using ANOVA and have updated the figure and the figure legend.

6) This data is also counter-intuitive based on figure 3g since suppression of RRM2 is associated with cell proliferation inhibition (PMID: 29749541) and E2F8 promotes proliferation (PMID: 26992224). How do the authors reconcile this discrepancy?

We appreciate this reviewer's comment and understand that these findings may have added confusion. The references provided by the reviewer suggest that one might expect decreased proliferation in LN given the decreased expression of RRM2 and E2F8 observed in AxLN sub-clones. However, in contrast to the article cited by the reviewer, the majority of scientific evidence instead points towards E2F7 and E2F8 acting as cell cycle

repressors, not promoters (Chen et al, *Nature Reviews Cancer*, 2009; PMID: 19851314). Thus, decreases in E2F7 and E2F8 would be consistent with increased proliferation. RRM2, on the other hand, has been shown to have pleiotropic functions beyond its role in the cell cycle, most notably in increasing invasiveness and metastasis. A large body of literature supports that increased RRM2 leads to a pro-migratory phenotype in many cancers (e.g. breast, gastric, colon, thyroid and hepatocellular carcinomas). Our working model is that HDAC11 suppression of E2F7 and E2F8 leads to increased proliferation and survival within the LN, and that loss of HDAC11 leads to de-repression of RRM2, which promotes an increased migratory phenotype from the LN to distant sites (Fig 6). The effect of HDAC11 on regulating proliferation and migration is undoubtedly the sum total effect of it acting on many genes in addition to E2F7, E2F8, and RRM2 that cannot be adequately addressed in a single study. In the discussion section we have further expounded on the context of our findings with what is known in the literature, as well as what is not known.

7) Figure 4b, again, ANOVA is the more appropriate statistical test. How do the authors reconcile the increase in E2F8 expression seen here with the decreased RRM2 and E2F8 expression seen with higher HDAC11 expression shown in figure 3h?

We have re-analyzed the data for Fig 4b using ANOVA. Please note this is now Fig 4a.

With regard to the prior Fig 4b (now Fig 4a) showing an increase in E2F8, among other targets (E2F7, RRM2, others), these findings are consistent with loss of HDAC11 (using 2 unique HDAC11 shRs) leading to de-repression of the targets. To help further clarify this directionality, we have illustrated in the graphical abstract that decreased HDAC11 results in increased acetylation of H3 and H4, thereby leading to increased gene expression of target genes.

8) ChIP-seq of HDAC11 in HEK-293T cells, which are kidney cells, may not represent relevant chromatin binding sites present in breast epithelial cells. Performing this in MBA-231 or some other breast cancer cell line would be more appropriate.

We appreciate this reviewer's comment, which is similar to Reviewer #2, Comment #6. We agree that HDAC11 over-expression in 293 cells is limited and may not be fully representative of what happens in breast epithelial cells. We therefore have over-expressed HDAC11 in MDA-MB-231 cells and found that HDAC11 is highly enriched at the promoters of several downstream target genes (Fig 4b).

In summary, we were very pleased to note that the reviewers found our work interesting, and important, and we appreciate their constructive suggestions for extending the scope and detail of these studies. Thank you in advance for your thoughtful consideration. Please let me know if I can provide any further information.

Sincerely,

Chad V. Pecot, M.D.

Reviewers' comments:

Reviewer #1 (Remarks to the Author):

The revised submission adequately addressed the concerns and questions raised.

Scott Thomas, PhD

Reviewer #2 (Remarks to the Author):

After a careful reading of the re-submitted version and considering all the points address by the authors, this reviewer feels that the manuscript has substantially improved. Overall, the authors have provided reasonable explanations, clarified my points and streamlined certain conceptual points. In addition, an effort to extend the finding to an additional cell line and to extend and correlate mRNA-based observations to protein levels is satisfactory. In summary, the current manuscript reads better, more clearly and is backed up with more consistent and robust data.

Reviewer #3 (Remarks to the Author):

This is a revised manuscript describing the investigations of the role of the lymphatic route of dissemination for the formation of distant metastasis. Using rapid autopsy samples, the investigators performed phylogenetic analysis and interpreted the data to indicate that distant metastases likely arose from tumors in the draining lymph nodes. Using a microinjection technique, the investigators then examined the relative contribution of distant metastases from labeled cells implanted into the mammary fat pad or draining lymph node. The data from these experiments suggested that LN cells were more efficient at forming distant metastases. The investigators subsequently performed molecular analysis to identify, and then characterize, the role of Hdac11 in this process.

While there is some very interesting information presented in this manuscript, including the fact that Hdac inhibitors may increase distant metastasis from tumor cells lodged in lymph nodes, there are a number of issues with this study that are still of concern. These include:

SNV and clade analysis is not consistent with a “primary tumor-to-lymph node-distant met” timeline. For patients A24 and A30, the lymph node sample is more distantly related to the primary tumor than the distant metastases. This would be consistent with a primary-to-distant met-lymph node order, or seeding of the two sites from different subclones within the primary tumor. The direct microinjection into the draining lymph node clearly demonstrates that, at least in this model, that lymph node tumors can seed distant metastases, but the primary human data should not be over interpreted to directly support this possibility.

Extended figure 3b suggests that LN seeding of distant mets is not a frequent occurrence in the 4T1 model. Since no difference is observed in metastatic index in the LN resected mice, this result suggests that the primary route of dissemination and colonization is hematogenous, otherwise one would expect a significant reduction of distant metastases in these animals.

HDAC11 expression from the rapid autopsy samples is not consistent with the hypothesis that HDAC11 is driving LN progression. The LN samples are not significantly different from the primary tumor samples and the apparent trend toward significance is driven by a single sample (A30).

The methylation changes observed across the Hdac11 promoter appear to be pretty modest. Is this level of methylation change sufficient to account for the differences in expression observed?

The antibody specificity problem revealed by the investigators in the rebuttal calls into question the validity of the ChIP experiments, if they are using the same antibodies for the western blots. While it is not uncommon for antibodies to work for one application, but not another, validation for the specificity of the antibody would greatly increase the confidence of the results. Performing an IP with an epitope tagged version of HDAC11, followed by a western blot targeting the epitope would demonstrate that the antibody is at least capable of pulling down HDAC11. Otherwise, as the investigators demonstrated in the rebuttal material, it isn't clear what the antibodies are actually interacting with.

Pharmacological inhibitors are not specific for Hdac11. Therefore it is not possible to unambiguously ascribe differences in phenotypes after treatment to a single histone deacetylase. Effects could be the result of targeting other deacetylases, or a combination of them.

Chad V. Pecot, M.D.
Assistant Professor

UNC Lineberger Comprehensive Cancer Center
Thoracic Medical Oncology
450 West Drive, Office 32.048
Chapel Hill, NC 27599
Office: 919-966-4779, Cell: 615-305-3955
pecot@email.unc.edu

August 1st, 2019

Dear Reviewers:

On behalf of my co-authors, I thank the referees for the review of our manuscript entitled, "Histone deacetylase 11 inhibition promotes breast cancer metastasis from lymph nodes" (NCOMMS-18-33493B) to be considered for publication in *Nature Communications*. We are deeply appreciative of the reviewers' comments and believe the subsequent experiments we have performed substantially improved the scope and impact of our manuscript. The purpose of this letter is to highlight all of these changes to the manuscript, and I have outlined these below:

We were very pleased that Reviewers #1 and #2 felt our revised manuscript "adequately addressed the concerns and questions raised" and was "substantially improved... provided reasonable explanations and streamlined certain conceptual points".

Response to Reviewer #3:

SNV and clade analysis is not consistent with a "primary tumor-to-lymph node-distant met" timeline. For patients A24 and A30, the lymph node sample is more distantly related to the primary tumor than the distant metastases. This would be consistent with a primary-to-distant met-lymph node order, or seeding of the two sites from different subclones within the primary tumor. The direct microinjection into the draining lymph node clearly demonstrates that, at least in this model, that lymph node tumors can seed distant metastases, but the primary human data should not be over interpreted to directly support this possibility.

We are assuming the reviewer is referring to patients A20 and A34 in Figure 1. We agree that for the clade analysis, a primary to distant metastasis to LN metastasis order of spread is also a possible interpretation of this data. We have noted this possibility in the manuscript.

Extended figure 3b suggests that LN seeding of distant mets is not a frequent occurrence in the 4T1 model. Since no difference is observed in metastatic index in the LN resected mice, this result suggests that the primary route of dissemination and colonization is hematogenous, otherwise one would expect a significant reduction of distant metastases in these animals.

We agree with the reviewer that this experiment shows that both routes of spread contribute to distant metastases, and that hematogenous metastasis may be the predominant route of spread. However, our findings are still highly clinically relevant. We demonstrate that although metastasis via the lymphatic route may be less frequent, it is also a more efficient process than the hematogenous route. Our manuscript focuses on addressing the significant knowledge gap regarding the mechanisms of metastasis from lymph node and does not discount that hematogenous dissemination contributes significantly to distant metastasis. We have further clarified this in the manuscript.

HDAC11 expression from the rapid autopsy samples is not consistent with the hypothesis that HDAC11 is driving LN progression. The LN samples are not significantly different from the primary tumor samples and the apparent trend toward significance is driven by a single sample (A30).

The data regarding expression of HDAC11 expression in patient samples is limited by the small sample size and will require further validation in a large cohort of clinical samples, which is outside the scope of this study. We have toned down conclusions that can be drawn from this limited dataset in the manuscript.

The methylation changes observed across the Hdac11 promoter appear to be pretty modest. Is this level of methylation change sufficient to account for the differences in expression observed?

Previous studies have shown that a difference in methylation at a single site within a CpG island is sufficient to alter the binding of transcription factors and affect gene transcription (Watt and Molloy, *Genes and Development*, 1988). The degree of methylation changes may also be sufficient to impact the physical properties of DNA that affect chromatin configuration and nucleosome assembly. Regardless, we agree that the DNA methylation results are correlative and cannot rule out the possibility that other factors may account for the differences in HDAC11 expression observed. We have edited the manuscript to reflect that this finding is correlative and that other mechanisms may also be involved.

The antibody specificity problem revealed by the investigators in the rebuttal calls into question the validity of the ChIP experiments, if they are using the same antibodies for the western blots. While it is not uncommon for antibodies to work for one application, but not another, validation for the specificity of the antibody would greatly increase the confidence of the results. Performing an IP with an epitope tagged version of HDAC11, followed by a western blot targeting the epitope would demonstrate that the antibody is at least capable of pulling down HDAC11. Otherwise, as the investigators demonstrated in the rebuttal material, it isn't clear what the antibodies are actually interacting with.

We thank the reviewer for the suggestion to validate the specificity of the HDAC11 antibodies using IP. We transfected 293T cells with empty vector (pcDNA) or HDAC11-FLAG ORF. Of note, these are the same constructs used in the HDAC11 over-expression experiments in the manuscript. We then performed an IP using IgG or the two HDAC11 antibodies that were used for the ChIP experiments. To test for the specificity of the HDAC11 antibodies, an immunoblot using an antibody against FLAG was then performed on the eluent from the IP beads. As shown in the figure, a band consistent with the molecular weight of HDAC11 is visible only in lane 4, which consists of 293T cells that have been transfected with HDAC11-FLAG and then undergone IP with two anti-HDAC11 antibodies.

Immunoblot for the FLAG epitope after immunoprecipitation with two HDAC11 antibodies demonstrates that the antibodies are capable of pulling down HDAC11.

Pharmacological inhibitors are not specific for Hdac11. Therefore, it is not possible to unambiguously ascribe differences in phenotypes after treatment to a single histone deacetylase. Effects could be the result of targeting other deacetylases, or a combination of them.

We agree with the critique that the pharmacologic inhibitors are not entirely specific for HDAC11. While it is likely that other HDACs are being affected by the HDAC inhibitors, the fact that the phenotypes mirror the shHDAC11 experiments suggest that the phenotypes are likely at least in part HDAC11-mediated. We have addressed this in the manuscript.

In summary, we were very pleased to note that reviewers #1 and #2 were fully satisfied with our revised manuscript. We have performed the final experiment requested by reviewer #3, which showed that the use of two anti-HDAC11 antibodies together in IP/ChIP is capable of demonstrating HDAC11-specificity, despite our concerns regarding the use of these antibodies for immunoblotting. Finally, we appreciate the opportunity to further clarify some of our key findings and have edited the manuscript so as to not overstate our conclusions. We believe that these revisions have further improved our manuscript.

Sincerely,

Chad V. Pecot, M.D.

REVIEWERS' COMMENTS:

Reviewer #3 (Remarks to the Author):

The authors have addressed all of my concerns. This manuscript is acceptable for publication in its current form.